# OVERTONE: CYCLIC PATCH MODULATION FOR CLEAN, EFFICIENT, AND FLEXIBLE PHYSICS EMULATORS

**Payel Mukhopadhyay**[1,3]**, Michael McCabe**[2,3]**, Ruben Ohana**[2,3]**, Miles Cranmer**[1,3]
[1]University of Cambridge, UK
[2]Flatiron Institute, New York, USA
[3]Polymathic AI

## ABSTRACT

Transformer-based PDE surrogates achieve remarkable performance but face two key challenges: fixed patch sizes cause systematic error accumulation at harmonic frequencies, and computational costs remain inflexible regardless of problem complexity or available resources. We introduce Overtone, a unified solution through dynamic patch size control at inference. Overtone's key insight is that cyclically modulating patch sizes during autoregressive rollouts distributes errors across the frequency spectrum, mitigating the systematic harmonic artifact accumulation that plague fixed-patch models. We implement this through two architecture-agnostic modules—CSM (using dynamic stride modulation) and CKM (using dynamic kernel resizing)—that together provide both harmonic mitigation and compute-adaptive deployment. This flexible tokenization lets users trade accuracy for speed dynamically based on computational constraints, and the cyclic rollout strategy yields up to 40% lower long rollout error in variance-normalised RMSE (VRMSE) compared to conventional, static-patch surrogates. Across challenging 2D and 3D PDE benchmarks, one Overtone model matches or exceeds fixed-patch baselines across inference compute budgets, when trained under a fixed total training budget setting. Code is made available at: https://github.com/payelmuk150/patch-modulator.

## 1 INTRODUCTION

Numerical solvers have long been the gold standard for simulating spatio-temporal dynamics in systems governed by partial differential equations (PDEs). Numerical solvers offer convergence guarantees, exploit problem sparsity, and provide fine-grained control of the accuracy-compute trade-off through configurable resolution (Morton & Mayers, 2005; Malalasekera, 2007), driven by both scientific necessity and computational availability (Berger & Colella, 1989; Wedi, 2014).

Deep learning has established itself as a powerful framework for training surrogate models of physical simulations (Cao, 2021; Li et al., 2023; McCabe et al., 2023; Herde et al., 2024). Though expensive to train, relative cheap inference costs result in overall compute savings, which has led deep learning surrogates to become popular in forecasting (Bi et al., 2023), PDE-constrained optimization (Li et al., 2022), and parameter inference (Cranmer et al., 2020; Lemos et al., 2023). Recent approaches are often derived from computer vision methods such as Vision Transformers (ViT (Dosovitskiy et al., 2020)), which segment discretized fields into patches and then tokens. Non-overlapping patches are used as a means to reduce token count and the associated quadratic attention cost.

However, these surrogates face two major limitations. First, we discover that in autoregressive rollouts, fixed patch sizes cause systematic error accumulation at harmonic frequencies through temporal coherence. While patch artifacts themselves are well-known, we show that repeatedly hitting the same harmonics at every timestep creates a distinct accumulation pattern. When patches of size $k$ are used consistently during autoregressive rollouts, the artificial boundaries inject errors at wavenumbers $n/k$ where $n$ is an integer. These errors constructively interfere over time, creating pronounced spectral spikes (Figure 2) and visible grid-like artifacts. Second, fixed-patch models

---

*Corresponding Email: `pm858@cam.ac.uk`, `payelmukhopadhyay180@gmail.com`

have inflexible computational cost. In physical modeling, different applications have hard resolution thresholds for resolving features like shocks or wave-fronts (Boyd, 2001), yet smaller patches that improve accuracy (Dosovitskiy et al., 2020; Wang et al., 2025a) cannot be selected post-training.

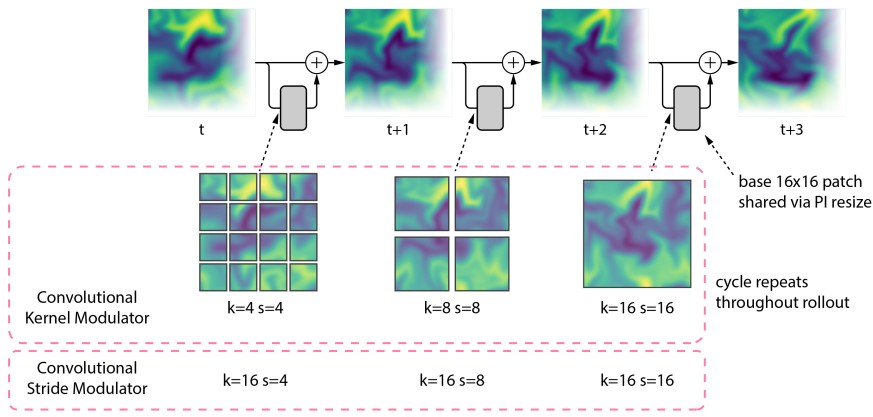

Figure 1: Illustration of Overtone's CSM (stride modulation) and CKM (kernel modulation). Cyclic modulation distributes errors across frequencies, preventing error accumulation at harmonics and enabling accuracy-compute trade-offs at inference time.

We address both challenges through dynamic patch size control during rollout—so far unexplored in autoregressive modeling. Cyclically alternating patch sizes between $k_1$, $k_2$, and $k_3$ at consecutive timesteps can mitigate artifacts from accumulating coherently at single harmonics, instead distributing them across the frequency spectrum. This same capability enables compute-adaptive deployment without retraining, allowing users to balance accuracy and computational demands based on available resources.

Overtone implements this through two modules: *Convolutional Stride Modulation (CSM)* maintains a static kernel but dynamically modulates stride, while *Convolutional Kernel Modulation (CKM)* uses bicubic interpolation to resize a single kernel (as in (Xu et al., 2014; Beyer et al., 2023)) for both encoder and decoder, all while cyclically modulating these during rollout (Figure 1). We empirically demonstrate that this cyclic modulation fundamentally alters artifact accumulation in autoregressive rollouts, also with motivation of the technique in spectral analysis (Figure 2), to show how temporal coherence in tokenization has been an unaddressed source of error in spatiotemporal models.

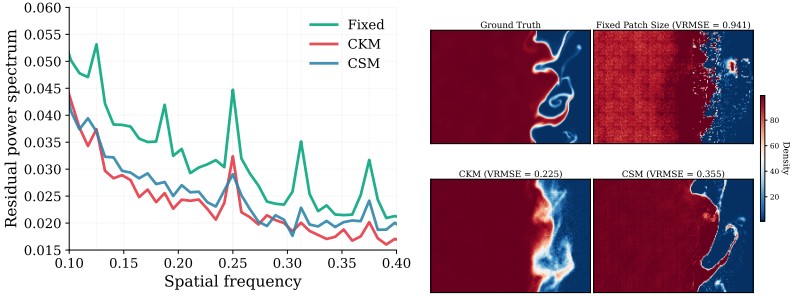

Figure 2: *Left:* Averaged 1D residual power spectrum at rollout step 20 revealing harmonic error accumulation in fixed-patch models. The fixed patch 16 model shows pronounced spikes at harmonic frequencies $k/16$, while CSM and CKM with cyclic modulation distribute spectral errors across the frequency range, supporting our theoretical motivation (Section 2). Note that this is residual spectrum wrt the ground truth, which is trivially zero for the true simulation, and is therefore not shown. *Right:* Spatial manifestation of harmonic artifacts at rollout step 20 on *Turbulent Radiative Layer 2D*. Cyclic patch modulation eliminates the grid-like distortions in the fixed model, showing the practical impact of Overtone's rollout strategy.

**Contributions.** We present Overtone, a framework that identifies and addresses a fundamental issue in patch-based surrogates while providing practical compute flexibility:

1. *Harmonic Artifact Diagnosis:* We identify that fixed-patch tokenization causes systematic spectral error accumulation at harmonic frequencies $k/p$, manifesting as spectral spikes and spatial artifacts that degrade long-term predictions in autoregressive surrogates.

2. *Cyclic Rollout Strategy:* We introduce cyclic stride/patch modulation during inference for autoregressive rollouts. While adaptive patching exists in vision, we are not aware of prior work that cyclically modulates patches during autoregressive rollouts in computer vision, video modeling, or PDE applications. The rollout cycles tokenization scale (e.g., $4 \rightarrow 8 \rightarrow 16$, repeated), which (i) prevents errors frequency space errors from repeatedly reinforcing at a single patch scale—thereby reducing patch-lattice harmonic artifacts—and (ii) periodically leverages finer tokenization (4 and 8) for higher-fidelity steps while retaining the efficiency of coarser patches (16). Together, these effects reduce VRMSE by up to 40% relative to fixed-patch baselines (patch size 16), *without retraining*.

3. *Architecture Agnostic Controllable Tokenization Modules:* We develop CSM (using dynamic stride adjustment) and CKM (using kernel interpolation (Xu et al., 2014; Beyer et al., 2023)) as architecture-agnostic methods for controllable tokenization. These modules operate at the tokenization level and are compatible with backbones like vanilla ViT Dosovitskiy et al. (2020), axial ViT Ho et al. (2020), CViT Wang et al. (2025b), making them flexible plug-ins for PDE surrogates.

4. *Compute-Adaptive Deployment:* Beyond strong accuracy gains, Overtone provides practical trade-offs between speed and precision at inference. A single model can adjust patch sizes based on available resources, matching or exceeding multiple fixed-patch baselines across all compute budgets on challenging 2D/3D PDE benchmarks from the Well (Ohana et al., 2024). Overtone is also competitive with a range of SOTA non-patch baselines, underscoring its broader significance.

We focus on autoregressive surrogate models for time-evolving PDE systems, where the goal is to predict the next state of a discretized physical field from a short temporal context.

## 1.1 BACKGROUND AND RELATED WORK

**Architecture-agnostic tokenization for PDE surrogates.** CSM and CKM are architecture-agnostic tokenization strategies applied at the encoder and decoder stages. They control the patch/stride configuration for spatial downsampling while remaining independent of the transformer architecture. We show compatibility with standard vanilla ViTs (full attention) (Dosovitskiy et al., 2020) and axial ViTs (Ho et al., 2019). This modularity makes CSM and CKM flexible plug-ins for diverse transformer-based PDE surrogates, allowing adaptive deployment without redesign.

**Surrogate modeling of PDEs with deep learning.** PINNs (Raissi et al., 2019; Cai et al., 2021; Han et al., 2018; Hao et al., 2023a; Karniadakis et al., 2021; Lu et al., 2021; Pang et al., 2019; Sun et al., 2020; Zhu et al., 2023) embed physics in loss functions, while CNN- and GNN-based surrogates operate on structured or unstructured meshes, supporting spatiotemporal tasks such as climate and fluid modeling (Brandstetter et al., 2022c; Gupta & Brandstetter, 2022; Janny et al., 2023; Li & Farimani, 2022; Lötzsch et al., 2022; Pfaff et al., 2021; Prantl et al., 2022; Sanchez-Gonzalez et al., 2020; Stachenfeld et al., 2022; Thuerey et al., 2020; Ummenhofer et al., 2020; Wang et al., 2020; Lam et al., 2022; Nguyen et al., 2023; Pathak et al., 2022). Neural operators (Li et al., 2020c;b; Lu et al., 2019; 2022; Ovadia et al., 2023; Brandstetter et al., 2022a;b; Cao, 2021; Gupta et al., 2021; Hao et al., 2023b; Jin et al., 2022; Kissas et al., 2022; Kovachki et al., 2021) aim for resolution-invariant learning, with FNOs (Li et al., 2020a) using Fourier transforms for global context.

**ViTs and compute adaptive models for PDE surrogate modeling.** Initially developed for image recognition tasks, ViTs (Dosovitskiy et al., 2020) have seen increasing adoption in the PDE surrogate modeling literature (Cao, 2021; Li et al., 2023; Liu et al., 2023). Despite their strengths, traditional ViTs rely on fixed patch sizes, which strongly limits their flexibility and adaptability when modeling PDE-based systems as discussed in the introduction. Examples of kernel interpolation for adaptive sizing include SiCNN (Xu et al., 2014), which applied bicubic interpolation to CNN kernels and trained at multiple scales, and FlexiViT (Beyer et al., 2023), which applies bicubic interpolation to

ViT kernels for image classification. Although explored in computer vision, adaptive patching for PDE surrogate modeling has only been explored in (Zhang et al., 2024a), the difference with our work being that their patchification is data-driven, while ours can be manually controlled based on compute requirements.

## 2 METHODS

**Overview.** CSM and CKM operate as encoder-decoder modules within a ViT based architecture for autoregressive PDE prediction. The transformer core applies a sequence of temporal attention, spatial attention, and MLP blocks (Appendix C.2). Crucially, Overtone's framework is architecture-agnostic—we test it with axial attention and vanilla ViTs (and also a recent model—Continuous ViT (Wang et al., 2025b) in Appendix E).

**Utility of flexible models.** As illustrated in Figure 3, increasing the number of tokens—equivalent to using smaller patch sizes in models with fixed patches—enhances the accuracy of autoregressive prediction tasks, but at a higher computational cost. The cost for models trained with fixed patch sizes cannot be controlled at inference time. This limits the utility of these models in budget-aware production settings and motivate the exploration of flexible models for PDE modeling with ViT-inspired architectures.

Research in the broader machine learning community has extensively demonstrated the value of scale (Zhai et al., 2022; Hoffmann et al., 2022). As transformer-based surrogate models similarly scale in parameter count, ensuring inference-time scalability is crucial, particularly to ensure these models are able to be used in compute-constrained environments. To avoid any confusion, by inference scalability, we mean patching/striding scalability or tunability, and not scalability in mesh resolution. Training and maintaining multiple fixed-patch models is a burden especially as large pretrained or foundation models are being built (McCabe et al., 2023; Herde et al., 2024; Morel et al., 2025). In such a scenario, if Overtone's flexible strategies can enable training of a *single* model without retraining for multiple patch resolutions without any loss in accuracy, these strategies can form the building blocks for larger scale foundation models for PDEs.

**Patching.** Given an input image with height $H$ and width $W$, ViTs will divide the image into non-overlapping patches of size $k \times k$, producing $N = \lfloor H/k \rfloor \cdot \lfloor W/k \rfloor$ tokens. For global attention, transformer cost scales as $\mathcal{O}(N^2)$, and for axial attention, as $\mathcal{O}(N\sqrt{N})$ (Ho et al., 2019). In convolutional tokenization, patches are extracted via kernels of size $k$ and stride $s$, resulting in token count $N = N_h \cdot N_w$ where $N_h = \lfloor (H - k)/s \rfloor + 1$ and $N_w = \lfloor (W - k)/s \rfloor + 1$. While most ViT based architectures use $k = s$, our CSM and CKM modules decouple these parameters to flexibly control token count at inference.

**Convolutional Stride Modulator (CSM).** CSM provides flexible downsampling by modulating the stride $s \in \{4, 8, 16\}$ at each forward pass. Consider input $x \in \mathcal{X}^T$ for some input space $\mathcal{X}^T \subseteq \mathbb{R}^{M \times H \times W \times T \times C}$, with $M, T$ and $C$ being the batch size, time context and input channels respectively. We pad the input with learned tokens (informed by boundary conditions) to avoid edge artifacts. During training, the model is trained with random stride sizes sampled from 4, 8, 16 during the forward pass (Appendix K.5). At inference, starting from input context $\hat{x}^0 = (x^1, ..., x^T)$, CSM uses a fixed kernel $w^{\text{base}}$ and computes:

$$x_{\text{enc}}^i = \underset{\text{stride } s_i}{\text{Conv}}(\hat{x}^i, w^{\text{base}}), \quad x_{\text{lat}}^i = \text{Processor}(x_{\text{enc}}^i), \quad \hat{x}_T^{i+1} = \underset{\text{stride } s_i}{\text{Conv}^{\mathsf{T}}}(x_{\text{lat}}^i, w^{\text{base}}),$$

where we cycle $s_i = (4, 8, 16)_{(i \bmod 3)+1}$ and slide $\hat{x}_{1:T-1}^{i+1} = \hat{x}_{2:T}^i$.

Here, $\text{Processor}$ denotes a transformer-based processor that operates on the encoded tokens, $\text{Conv}^{\mathsf{T}}$ indicates a transposed convolution, and the output $\hat{x}_T^{i+1} \in \mathcal{X}^1 \subseteq \mathbb{R}^{B \times H \times W \times 1 \times C}$ is the next-step prediction. CSM is compatible with both single-stage and multi-stage encoder-decoder modules. Implementation details and code are provided in the supplementary materials.

**Convolutional Kernel Modulator (CKM).** CKM introduces dynamic patch size selection within convolutional encoder and decoder blocks, enabling models to select a patch size $k \in \{4, 8, 16\}$ for each forward pass—powers-of-two that reflect standard discretizations in PDE data. Unlike most ViT-based surrogates that fix patch size at 16 (McCabe et al., 2023; Morel et al., 2025), CKM provides flexible tokenization essential for harmonic error mitigation (analyzed in Section 2). This is achieved using kernel interpolation (Xu et al., 2014; Beyer et al., 2023).

At training time, the model sees a random kernel size sampled from 4, 8, 16 (Appendix K.6). At inference, starting from $\hat{x}^0 = (x^1, ..., x^T)$, CKM resizes a base convolutional kernel $w^{\text{base}} \in \mathbb{R}^{k_{\text{base}} \times k_{\text{base}} \times C \times C'}$ using PI-resize (detailed in Appendix K) and computes:

$$x_{\text{enc}}^i = \underset{\text{stride } k_i}{\text{Conv}}(\hat{x}^i, B_{k_i}^{\mathsf{T}\dagger} w^{\text{base}}), \quad x_{\text{lat}}^i = \text{Processor}(x_{\text{enc}}^i), \quad \hat{x}_T^{i+1} = \underset{\text{stride } k_i}{\text{Conv}}^{\mathsf{T}}(x_{\text{lat}}^i, B_{k_i}^{\mathsf{T}\dagger} w^{\text{base}}),$$

where we cycle $k_i = (4, 8, 16)_{(i \bmod 3)+1}$ and slide $\hat{x}_{1:T-1}^{i+1} = \hat{x}_{2:T}^i$.

Here, $B_{k_i} \in \mathbb{R}^{k_i \times k_{\text{base}}}$ is a bicubic interpolation matrix and $B_{k_i}^{\mathsf{T}\dagger}$ its pseudoinverse transpose.

Examples of kernel interpolation for adaptive sizing include SiCNN (Xu et al., 2014), which applied bicubic interpolation to CNN kernels and trained at multiple scales, and FlexiViT (Beyer et al., 2023), which applies bicubic interpolation to ViT kernels for image classification. Using kernel interpolation, we introduce cyclic modulation during rollouts—transforming it from a compute-flexibility tool into a method for mitigating harmonic error accumulation in long-horizon predictions. In our vanilla ViT and Axial ViT models, we adopt two-stage convolutional encoders and decoders following the hMLP architecture (Touvron et al., 2022), which has recently been adopted for large-scale PDE surrogates (Morel et al., 2025; McCabe et al., 2023), and apply CKM independently at each stage.

Together, the Overtone framework offers general strategies for flexifying convolutional tokenization in ViT-based autoregressive schemes, as well as mitigating accumulating patch artifacts.

**Cyclic rollout strategy.** In standard autoregressive forecasting, patch sizes remain fixed across the rollout. By contrast, Overtone's flexible modules enable patch sizes or strides to alternate across timesteps, introducing new temporal patterns in tokenization. This test-time capability, decoupled from training, leads to a striking empirical effect: alternating rollouts *consistently suppress patch artifacts*—periodic errors that appear as harmonics in spectral residuals (Figure 2)

These artifacts are ubiquitous in fixed-patch ViT based models, regardless of attention type (vanilla, axial, Swin), and cannot be removed through training alone (Appendix F). Alternating rollouts yield cleaner, more stable long-horizon predictions. Our findings are primarily empirical, though we provide a mathematical motivation for this phenomenon below. This discovery was only possible due to the test-time flexibility of Overtone—a capability entirely absent from prior PDE surrogates.

**Why cycling mitigates artifacts.** Under the simplifying assumption of a linearized error model, the evolution can be written as $e_{n+1}(\omega) = \lambda(\omega)e_n(\omega) + a_n(\omega)$, where $\lambda$ propagates existing errors and $a_n$ injects fresh errors at patch boundaries. With a fixed patch size $k$, these injections align at harmonic frequencies $m/k$, remaining phase-locked across timesteps and leading to rapidly compounding error—visible as spectral spikes in Figure 2. By contrast, varying $k$ across timesteps disrupts this temporal coherence, spreading errors more evenly and mitigating growth. This should be understood as a heuristic explanation consistent with our empirical findings. See Appendix B for a more extended mathematical sketch and physical intuition.

## 3 RESULTS

To evaluate the practical advantages of compute-adaptive inference and validate the harmonic mitigation benefits, we conduct experiments on 2D and 3D datasets from The Well (Ohana et al., 2024). Our evaluation shows how a single flexible model can serve multiple deployment scenarios while also achieving superior accuracy through harmonic error distribution.

**Experimental setup.** We compare training a *single* flexible model versus *multiple* fixed-patch models. We train three fixed-patch models (patch sizes 4, 8, 16) for a fixed number of optimizer steps each, while training CSM and CKM models once with the same total compute budget. See Appendix C.3 for details.

This setup reflects a practical question for enabling adaptive compute: given a fixed training budget, is it better to train one flexible model or several static ones? This corresponds to deployment settings where inference budgets may vary (or are unknown a priori) and a single model must support multiple compute points; when only a single tokenization is required, training a specialized fixed-patch model remains a natural choice. All evaluations are performed on models trained under this constraint. After training, we compare the performance of CSM and CKM models against individual models trained at static patch sizes. If the compute-elastic models can enable flexibility without incurring a significant accuracy loss, this supports deploying a single flexible PDE surrogate across multiple inference-time compute settings without training and maintaining separate fixed-patch models.

We test on both axial ViT (Ho et al., 2019) and vanilla ViT (Dosovitskiy et al., 2020) architectures to measure generality. The input consists of six time frames, with loss optimized for next-step prediction. We use this teacher forced training setup throughout, which matches The Well setup and the majority of PDE surrogates we compare against. This ensures a fair comparison across all baselines. Alternating training strategies, such as rollout training, is complementary to our work and can additionally help to improve prediction accuracy further for *all* models. Incorporating them alongside Overtone is an interesting direction for future work. See Appendix C.1 for dataset details and Appendix I for ablations.

## 3.1 FLEXIBLE INFERENCE

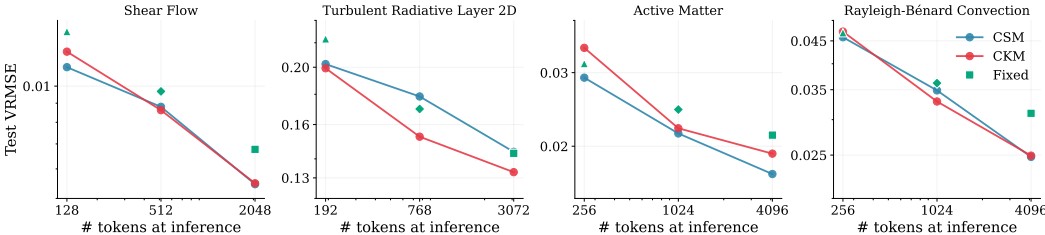

Figure 3: Next-step prediction test VRMSEs v.s. number of tokens at inference of a 100M parameter model trained on four 2D datasets. Lower VRMSE is better. Token count is a proxy for required compute. Note that there are three separate fixed-patch models (*green*), as each needs to be trained from scratch. This plot shows the compute-accuracy trade-off, which is also explored more in Section 3.1.

We choose the VRMSE metric (Appendix C.4) as a measure of accuracy. In Table 1 and Figure 3, we report VRMSE across a range of 2D and 3D PDE datasets from the Well (Ohana et al., 2024). Each model is evaluated at multiple token counts, which correspond to varying stride sizes in CSM or patch sizes in CKM—revealing the impact of inference-time compression on predictive accuracy. Importantly, we train the fixed patch size (p.s.) models as separate networks at each patch size, while we train CSM/CKM only once using random patch/stride schedules and evaluated without retraining. We show results for two different transformer architectures, Axial ViT (Ho et al., 2019) and vanilla full attention ViT (Dosovitskiy et al., 2020). Note that the goal is not to compare the two base architectures against each other, but to show that for both architectures, higher token count implies greater accuracy, and that CSM/CKM can provide a token count tunability at inference against the corresponding fixed patch baselines *without* losing significant accuracy.

Across all datasets and token configurations, CSM and CKM consistently match or outperform their fixed-patch counterparts—despite being single, flexible models. This holds for both axial and vanilla attention processors. Increasing the number of tokens at inference—by using smaller strides (CSM) or patch sizes (CKM)—improves VRMSE but increases compute (Table 1, Figure 3). For example, in the *Active Matter* dataset ($256^2$), reducing patch/stride size from 16 to 4 increases token count from 256 to 4096, triples inference time (0.21s/step to 0.63s/step), and increases compute from 5 to 170 GFLOPs—but reduces error by over 30%. Similarly, for 3D datasets ($64^3$), decreasing patch

size from 16 to 4 increases token count from 64 to 4096, increases inference time $8\times$ (0.11s/step to 0.8s/step), but improves VRMSE by $\sim 2\times$.

Token count directly controls the compute-accuracy trade-off. A single CSM/CKM model matches or outperforms multiple fixed models at every compute budget, without retraining. They excel across all frequency bands (BSNMSE analysis in Appendix D), preserving both coarse and fine features.

| Dataset | # Tokens | Vanilla ViT (100M) | | | Axial ViT (50M) | | | Well Baseline |
| | | CSM | CKM | f.p.s. | CSM | CKM | f.p.s. | |
|---|---|---|---|---|---|---|---|---|
| *Shear Flow* | 2048 | **0.00546** | 0.00549 | 0.00677 | 0.0077 | **0.0070** | 0.0085 | 0.1049 |
| | 512 | 0.0088 | **0.0086** | 0.0096 | 0.0104 | **0.0098** | 0.0124 | |
| | 128 | **0.0112** | 0.0124 | 0.0140 | **0.0135** | 0.0143 | 0.0173 | |
| *Turbulent Radiative Layer 2D* | 3072 | 0.146 | **0.133** | 0.143 | 0.153 | **0.145** | 0.153 | 0.2269 |
| | 768 | 0.179 | **0.154** | 0.173 | 0.179 | **0.165** | 0.186 | |
| | 192 | 0.204 | **0.200** | 0.225 | 0.211 | **0.209** | 0.238 | |
| *Active Matter* | 4096 | **0.0171** | 0.0192 | 0.0213 | **0.0238** | 0.0244 | 0.0276 | 0.0330 |
| | 1024 | **0.0214** | 0.0221 | 0.0245 | 0.0282 | **0.0260** | 0.0340 | |
| | 256 | **0.0294** | 0.0344 | 0.0315 | **0.0368** | 0.0369 | 0.0435 | |
| *Rayleigh-Bénard* | 4096 | **0.0248** | **0.0250** | 0.0310 | 0.0296 | **0.0291** | 0.0361 | 0.2240 |
| | 1024 | 0.0349 | **0.0328** | 0.0362 | 0.0406 | **0.0388** | 0.0443 | |
| | 256 | **0.0459** | 0.0467 | 0.0469 | **0.0537** | 0.0548 | 0.0582 | |
| *Supernova Explosion* | 4096 | 0.287 | **0.267** | 0.272 | 0.305 | 0.289 | **0.288** | 0.3063 |
| | 512 | 0.380 | **0.362** | 0.387 | 0.388 | **0.371** | 0.392 | |
| | 64 | 0.417 | **0.414** | 0.426 | 0.421 | **0.419** | 0.434 | |
| *Turbulence Gravity Cooling* | 4096 | 0.121 | **0.103** | 0.118 | 0.138 | **0.124** | 0.127 | 0.2096 |
| | 512 | 0.183 | **0.164** | 0.179 | 0.197 | **0.181** | 0.194 | |
| | 64 | **0.199** | 0.203 | 0.199 | 0.215 | 0.216 | **0.212** | |

Table 1: Test VRMSE across multiple 2D and 3D PDE datasets for next-step prediction using CSM, CKM, and fixed patch size (f.p.s.) models. We train CSM/CKM models once and evaluate them across multiple token counts (i.e., patch/stride configurations), whereas we train each fixed p.s. model separately. Results are shown for two transformer backbones: Axial ViT (50M) (Ho et al., 2019) and Vanilla ViT (100M) Dosovitskiy et al. (2020). We also report the best outcomes from the Well benchmark suite (Ohana et al., 2024), although after retraining the Well baselines for accurate results.

Table 1 includes Well benchmarks: FNO (Li et al., 2021), TFNO (Kossaifi et al., 2023), U-Net, and CNext-U-Net (Liu et al., 2022) (Ohana et al., 2024), re-trained with tuned rates for best accuracy. Our flexible models outperform these baselines across most tasks and budgets. We further demonstrate the modularity of our approach by integrating CKM with CViT (Wang et al., 2025b), a recent hybrid architecture. Results show flexified CViT consistently outperforms fixed-patch variants across four 2D PDE datasets. Full details are in Appendix E.

### 3.2 CSM & CKM ENABLE NEW INFERENCE-TIME STRIDE/PATCH SCHEDULES

A key advantage of compute-elastic stride/patch tokenization is that it enables *inference-time rollout schedules* that are inaccessible to fixed-tokenization patch-based surrogates. In standard patch-based models, inference repeatedly applies an identical patch at every autoregressive step. In contrast, compute-elastic models allow the stride and patch size to vary dynamically across rollout steps, without retraining. To our knowledge, PDE surrogates rarely treat inference time rollouts as a controllable knob. Here, compute-elastic tokenization turns rollout inference into a controllable scheduling problem, enabling budget- and horizon-aware rollouts without retraining.

Using a single trained compute-elastic model, we vary the inference-time stride/patch schedule during autoregressive prediction. We take the fixed patch=16 model (trained with the setup described in 3)

as a natural fixed-tokenization reference point, and compare it against the CSM/CKM alternating schedule that cycles stride and patch sizes over time at inference. Quantitatively, Table 2 shows rollout VRMSE averaged over 10 steps. Across both processor backbones, flexible models consistently outperform the fixed patch models—often by large margins—demonstrating greater rollout stability.

| Dataset | Axial ViT (50M) | | | | Vanilla ViT (100M) | | | |
|---|---|---|---|---|---|---|---|---|
| | CSM | CKM | f.p.s. | Δ | CSM | CKM | f.p.s. | Δ |
| *Shear Flow* | **0.0565** | 0.0579 | 0.0793 | +28.7% | **0.0375** | 0.0385 | 0.0504 | +25.6% |
| *TRL-2D* | **0.475** | 0.570 | 0.571 | +16.8% | **0.373** | 0.409 | 0.446 | +16.4% |
| *Active Matter* | **0.384** | 0.390 | 0.640 | +40.0% | 0.359 | **0.351** | 0.370 | +5.1% |
| *Rayleigh-Bénard* | **0.166** | 0.2159 | 0.217 | +23.5% | **0.140** | 0.215 | 0.2273 | +38.4% |
| *Supernova* | 1.310 | **1.270** | 1.905 | +33.3% | 1.20 | **1.14** | 1.75 | +34.9% |
| *TGC* | **0.667** | 0.680 | 0.860 | +22.4% | 0.559 | **0.527** | 0.77 | +31.6% |

Table 2: 10-step averaged rollout VRMSE across datasets. CSM/CKM vs. fixed patch size (f.p.s) models. Δ shows percentage improvement of best flexible model vs f.p.s. Abbreviations: *TRL-2D*: *Turbulent Radiative Layer 2D*, *TGC*: *Turbulence Gravity Cooling*, *Supernova*: *Supernova Explosion*.

Table 3 compares representative inference-time stride schedules that use the same set of strides $\{4, 8, 16\}$ but differ in their ordering over the rollout, and we find that the $4 \to 8 \to 16$ ordering performs best among the tested permutations. All permutations use the same multiset of strides and therefore have the same total token count over the rollout (for the random case, its the same total token count on average throughout the rollout with uniform sampling); they differ only in temporal ordering. These results show that beyond the overall compute budget, the timing of when higher-resolution tokenization is applied materially affects the rollout stability.

We also evaluate additional schedule families beyond permutations of the same cycle (Appendix A). These include (i) a two-step dwell periodic schedule that holds each stride/patch value for multiple consecutive steps, (ii) warm-up schedules that use smaller stride/patch values early in the rollout before switching to larger values, and (iii) schedules that use larger stride/patch values for most steps with occasional smaller-value steps. Across representative systems, some of these alternatives achieve rollout error close to the $4 \to 8 \to 16$ baseline, while others degrade substantially, reinforcing that schedule choice is a novel inference-time control knob enabled by compute-elastic tokenization.

| Dataset | 4→8→16 | 8→4→16 | 16→4→8 | Random |
|---|---|---|---|---|
| *Shear Flow* | **0.0375** | 0.0442 (−17.8%) | 0.0437 (−16.5%) | 0.0433 (−15.5%) |
| *Supernova Explosion* | **1.204** | 1.290 (−7.1%) | 1.325 (−10.0%) | 1.369 (−13.7%) |
| *Turbulence Gravity Cooling* | **0.5592** | 0.606 (−8.4%) | 0.6101 (−9.1%) | 0.6136 (−9.7%) |

Table 3: Vanilla ViT (CSM): 10–step average rollout VRMSE for representative inference-time stride/patch schedules. Percentage values indicate relative degradation in VRMSE compared to the $4 \to 8 \to 16$ baseline.

## 3.3 HARMONIC ARTIFACT MITIGATION

Our theoretical motivation (Section 2) suggests cyclic patch modulation distributes errors across frequencies, preventing accumulation at harmonics. We validate this through alternating rollout: CSM and CKM models cycle through stride/patch sizes of 4, 8, and 16 at each prediction step, avoiding the *harmonic artifacts* at frequencies $k/p$ that plague all fixed-patch ViT variants (vanilla, axial, Swin) and arise from tokenization mechanics, not training (Appendix F).

Figures 2, 4 and 5 illustrate this behavior for representative rollouts from *Turbulent Radiative Layer 2D*, *Active Matter* and *Shear Flow*. In each case, fixed p=16 inference exhibits growing spatial artifacts at long horizons, while scheduled rollouts remain visually smooth and physically coherent. We identify that this effect is prominently reflected in the residual power spectra of each of the figures, where scheduled inference reduces error power at the patch-lattice frequencies. In particular,

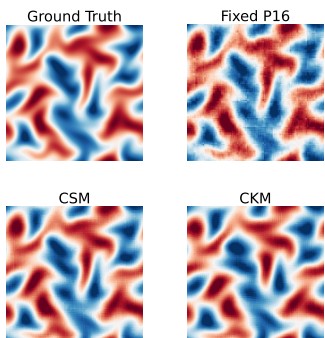
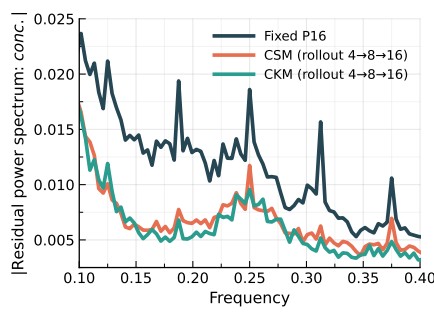

Figure 4: Step 20 rollout for the concentration field of the *Active Matter* dataset. **Left:** ground truth, fixed p=16, CSM, and CKM. **Right:** corresponding 1D averaged residual frequency spectrum.

under fixed-patch inference the residual spectra exhibit pronounced peaks at patch-lattice harmonic wavenumbers proportional to $k/p$ (for integer $k$), consistent with periodicity induced by the patch grid, and these peaks are substantially attenuated under cyclic rollout inference. Additionally, our models respect physics constraints over long rollouts, indicating they are not just memorizing patterns (See Appendix H for shear flow long trajectory rollout and physics based validation). See Appendix G for additional visualizations. These improvements emerge *without additional training*, achieved purely through inference-time patch modulation.

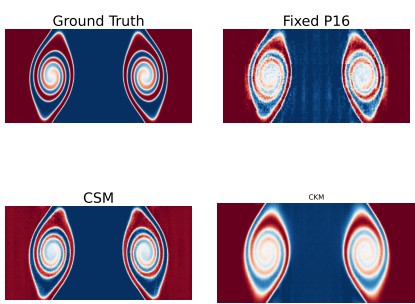
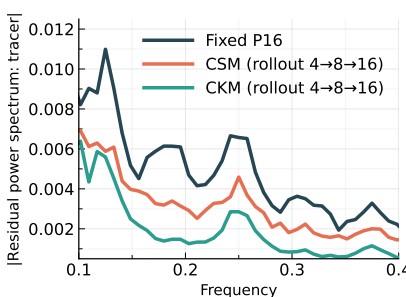

Figure 5: Step 40 rollout for the tracer field of the *Shear Flow* dataset. **Left:** ground truth, fixed p=16, CSM, and CKM. **Right:** corresponding residual frequency spectrum.

### 3.4 COMPARISON TO NON-PATCHED BASELINES

Although our focus is not on non-patch architectures, Table 4 and Table 11 provides a reference comparison to state-of-the-art CNN/Neural-Operator surrogates. It shows that flexible models are competitive with SOTA PDE surrogate architectures that do not use patch-based tokenization (e.g., TFNO (Kossaifi et al., 2023), FFNO (Tran et al., 2023), Transolver (Wu et al., 2024), SineNet (Zhang et al., 2024b)). Overtone's flexible models often significantly outperform these baselines. This result reinforces the potential of adaptive patch-based transformers to serve as competitive surrogates for PDE modeling, without sacrificing accuracy or flexibility. We did this architectural comparison for two complex physical systems, *Shear Flow* and *Turbulent Radiative Layer 2D* from the Well (Ohana et al., 2024) (comparisons on two more datasets in Table 11).

### 3.5 ABLATIONS

Ablations across model sizes (7M-100M) confirm: smaller patches improve accuracy, single flexible models outperform multiple fixed ones (within our total budget matched setting), and inference adapts without retraining (Appendix I.1). The approach remains robust to additional patch options (Appendix I.2), temporal context (Appendix I.4), and underlying base patch size (Appendix I.3). Reducing patch diversity during training (e.g., omitting $P = 8$) degrades performance at the excluded

Table 4: 10-step rollout performance comparison with non-patch based PDE surrogates on *Shear Flow* and *Turbulent Radiative Layer 2D* datasets. Our flexible models show competitive performance while offering compute adaptability.

| Model→ | TFNO | FFNO | SineNet | Transolver | CSM + ViT (Ours) | | CKM + ViT (Ours) | |
| --- | --- | --- | --- | --- | --- | --- | --- | --- |
| # Parameters→ | 20M | 6.2M | 35M | 11M | 7M | 100M | 7M | 100M |
| *Shear Flow* | 0.89 | 0.105 | 0.17 | > 1 | 0.076 | **0.0375** | 0.096 | 0.0385 |
| *Turbulent Radiative Layer 2D* | 0.81 | 0.485 | 0.65 | > 1 | 0.458 | **0.373** | 0.477 | 0.409 |

size, underscoring the value of patch variety (Appendix I.5). This ablation notes that we do not claim zero-shot generalization to unseen patch/stride configurations.

**Limitations.** Our approach requires training with multiple patch sizes to enable inference-time flexibility—we do not claim zero-shot adaptation to unseen patch configurations. This is a deliberate design choice rather than a limitation: training with diverse patch sizes ensures robustness across the operating range. Zero-shot patch adaptation remains an open problem for future work. Additionally, as is standard for models at the 50M–100M parameter scale, our experiments use single seeds due to computational constraints. This practice is consistent with recent large-scale PDE models (MPP, DISCO, CViT, Poseidon) which face similar resource limitations. The breadth of our experiments across multiple datasets and architectures helps compensate for the lack of error bars. Finally, the results are for regular grids, and extension to irregular geometries remains open for future work.

## 4 CONCLUSION

We presented Overtone, a unified framework for compute-adaptive inference in transformer-based PDE surrogates that addresses both practical deployment challenges and fundamental accuracy limitations. Overtone provides dynamic patch size control at inference time without retraining, allowing users to balance computational resources and accuracy requirements on demand. This flexibility solves a key problem in production deployment where different applications have varying computational budgets and accuracy needs.

Beyond practical compute flexibility, we discovered that cyclic patch modulation fundamentally improves rollout stability by mitigating harmonic artifacts. Fixed patch sizes cause errors that interfere constructively at frequencies $k/p$, manifesting as spectral spikes and grid artifacts. By alternating patch sizes during rollouts, we distribute these errors across the frequency spectrum, preventing coherent accumulation—reducing patch based artifacts compared to fixed baselines.

To implement these capabilities, we developed two architecture-agnostic modules for Overtone: CSM, which uses dynamic stride modulation, and CKM, which uses kernel interpolation for dynamic kernel control. Across challenging 2D and 3D benchmarks, a single Overtone model matches or exceeds multiple fixed-patch baselines across all compute budgets. This dual benefit—practical deployment flexibility plus core accuracy improvement—makes Overtone valuable for both research and production settings where computational efficiency and prediction quality are paramount.

**Future directions.** The harmonic error distribution idea extends beyond PDE surrogates to any autoregressive, vision-based model. Future work could explore adaptive (rather than cyclic) modulation strategies, extend these techniques to patch-based methods beyond transformers (Appendix J), and apply the same insights to video prediction and other spatiotemporal tasks. Additionally, our strategy could be applied to other ViT variants like Swin Liu et al. (2021) or CSWin Dong et al. (2022).

Finally, CSM/CKM could be integrated into large foundation models for physical systems (McCabe et al., 2023; Herde et al., 2024; Hao et al., 2024). In such settings, inference-time stride/patch control may enable flexible adaptation across downstream tasks that naturally occupy different points along the compute–accuracy spectrum. Recent work has, in fact demonstrated the effectiveness of this approach at scale by incorporating CSM into a large multiphysics foundation model, *Walrus*, yielding strong performance across a wide range of 2D and 3D systems (McCabe et al., 2025). Together, these results position flexible tokenization as a key tool for improving both computational efficiency and long-horizon accuracy in autoregressive surrogates.

ACKNOWLEDGEMENTS

Polymathic AI gratefully acknowledges funding from the Simons Foundation and Schmidt Sciences, LLC. Payel Mukhopadhyay thanks the Infosys-Cambridge AI centre for support. The compute for this project involved multiple sources, and the authors would like to thank the Scientific Computing Core, a division of the Flatiron Institute, a division of the Simons Foundation for extensive computational support. Miles Cranmer is grateful for support from the Schmidt Sciences AI2050 Early Career Fellowship and the Isaac Newton Trust. We thank Geraud Krawezik for help with compute resources, and the broader Polymathic AI team for valuable discussions and feedback.

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

| Dataset | $4 \to 8 \to 16$ | coarse+bursts | two-step dwell | warm-up→coarse | reverse warm-up |
|---|---|---|---|---|---|
| *Shear Flow* | 0.0375 | 0.0514 $(-37.1\%)$ | 0.0380 $(-1.3\%)$ | 0.03837 $(-2.3\%)$ | 0.05407 $(-44.2\%)$ |
| *Supernova Explosion* | 1.204 | 1.533 $(-27.3\%)$ | 1.208 $(-0.3\%)$ | 1.212 $(-0.7\%)$ | 1.5999 $(-32.9\%)$ |
| *Turbulence Gravity Cooling* | 0.5592 | 0.689 $(-23.2\%)$ | 0.5506 $(+1.5\%)$ | 0.5577 $(+0.3\%)$ | 0.724 $(-29.5\%)$ |

Table 5: Vanilla ViT (CSM): 10–step average rollout VRMSE for additional inference-time schedule families. Percentages indicate relative change compared to the $4 \to 8 \to 16$ baseline (positive is better / lower VRMSE compared to $4 \to 8 \to 16$ baseline ). All results use the same trained model as Table 3; only the inference-time stride/patch schedule differs.

## A    ADDITIONAL INFERENCE-TIME SCHEDULE FAMILIES

To complement the permutation study in Table 3, we evaluate several additional families of inference-time stride/patch schedules designed to probe qualitatively different behaviors beyond cyclic or random orderings. The goal of this appendix experiment is not to optimize schedules, but to check whether the improvements reported for the $4 \to 8 \to 16$ cycle are specific to that particular periodic pattern, and to illustrate how different schedule "shapes" (e.g., bursty compute allocation or horizon-dependent policies) affect rollout accuracy.

All results in this section use the same trained Vanilla ViT (CSM) model as in Table 3; we vary only the inference-time stride/patch sequence. For clarity, we list the 10–step schedules explicitly (written as stride/patch per rollout step):

- **Baseline periodic (used in the main text):** $4 \to 8 \to 16$ (repeat), $[4, 8, 16, 4, 8, 16, 4, 8, 16, 4, .....]$.

- **Mostly coarse + bursts:** a bursty schedule that uses coarse tokenization for most steps with occasional fine "bursts", $[16, 16, 16, 16, 4, 16, 16, 16, 16, 4]$.

- **Two-step dwell periodic:** a periodic schedule with longer dwell time at each tokenization level (two consecutive steps per stride value), $4 \to 4 \to 8 \to 8 \to 16 \to 16$ (repeat), $[4, 4, 8, 8, 16, 16, 4, 4, 8, 8, ...]$.

- **Warm-up then coarse (early fine):** a horizon-dependent schedule that allocates finer strides early 4488 in the rollout and then switches to larger strides of 16 for the rest of the rollout, $[4, 4, 8, 8, 16, 16, 16, 16, 16, 16, ...]$.

- **Reverse warm-up (late fine):** the reverse horizon-dependent schedule, which delays smaller strides until late in the rollout, $[16, 16, 16, 16, 16, 16, 8, 8, 4, 4]$.

Table 5 supports two main observations. First, the rollout improvements from scheduling are not confined to a single periodic pattern: both the two-step dwell periodic schedule and the warm-up then coarse schedule achieve performance close to the $4 \to 8 \to 16$ baseline on these representative systems. This suggests that multiple simple schedule families can yield similar rollout VRMSEs when allocating tokenization resolution across rollout steps.

Second, schedule choice still matters: not all ways of varying tokenization across steps are beneficial. In particular, the mostly coarse + bursts schedule and the reverse warm-up schedule substantially degrade rollout accuracy on these systems. Together with the permutation results in Table 3, these results indicate that rollout-time tokenization is a meaningful inference-time control knob, while exploration of more sophisticated schedules like dataset dependent schedules, or choosing optimal scheduling automatically as rollout progresses is left for future research.

On the artifacts side, we find that although the warm-up then coarse schedule can achieve rollout error close to the $4 \to 8 \to 16$ baseline (Table 5), after the initial warm-up it uses a constant stride value (16) for the remainder of the rollout. Figure 6 shows that, in our experiments, schedules that alternate stride/patch values throughout the rollout (e.g., $4 \to 8 \to 16$ repeat and $4 \to 4 \to 8 \to 8 \to 16 \to 16$ repeat) were more effective at attenuating patch-lattice harmonic peaks in the residual spectrum than warm-up schedules that become constant later in the rollout. This highlights that schedule form affects artifact behavior beyond aggregate rollout VRMSE error.

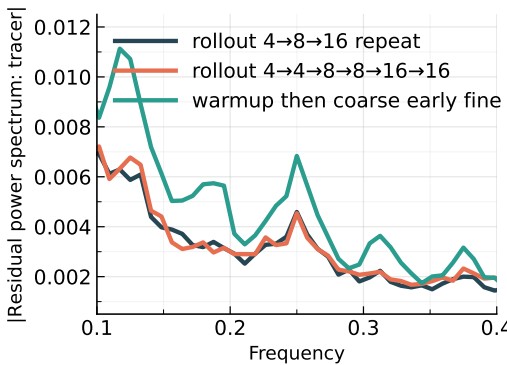

Figure 6: Averaged 1D residual power spectrum (tracer field, *Shear Flow*) for long-horizon rollouts under different inference-time stride/patch schedules: $4\to8\to16$ repeat, $4\to4\to8\to8\to16\to16$ repeat, and a warm-up then coarse (early fine) schedule that transitions from $4$ and $8$ to $16$.

## B  HARMONIC ERROR ANALYSIS

Consider the error between predicted and true fields during autoregressive rollouts. Working in the Fourier domain under a first-order linearization, let $e_n(\omega) \in \mathbb{C}$ denote the Fourier coefficient of error at step $n$, frequency $\omega \in [-\pi, \pi]^2$. The error evolves as:

$$e_{n+1}(\omega) = \lambda(\omega)e_n(\omega) + a_n(\omega), \tag{1}$$

where $\lambda(\omega) \in \mathbb{C}$ is the linearized propagation factor (how the model propagates existing errors) and $a_n(\omega)$ is the fresh error injected at step $n \geq 0$ by patch boundaries, which we assume $a_n$ is zero-mean with finite variance. Throughout this section, $\mathbb{E}[\cdot]$ denotes expectation over stochastic realizations of the injected errors, while $\langle\cdot\rangle_n$ denotes averaging over the rollout step index $n$.

A fixed patch grid of period $k$ acts as a spatial Dirac comb, concentrating error injection at the harmonic lattice:

$$\Omega_k := \left\{\omega = \left(\frac{2\pi m_x}{k}, \frac{2\pi m_y}{k}\right) : m_x, m_y \in \mathbb{Z}\right\}. \tag{2}$$

When cycling through patch sizes $\{k_1, k_2, k_3\}$, errors are injected on $\Omega_\cup = \bigcup_j \Omega_{k_j}$ (the union of all harmonic lattices). However, only frequencies in $\Omega_\cap = \bigcap_j \Omega_{k_j}$ (the intersection—frequencies common to all patch sizes) maintain consistent spatial alignment across all steps.

Solving the linear recurrence yields $e_n(\omega) = \sum_{i=1}^{n} \lambda(\omega)^{n-i} a_i(\omega)$. The key question is: do the injected errors $a_i(\omega)$ add up constructively or destructively? To quantify this, let $\sigma^2(\omega) = \mathbb{E}[|a_n(\omega)|^2]$ and define the normalized coherence between errors separated by $\tau$ steps:

$$\gamma_\tau(\omega) := \frac{\langle\mathbb{E}[a_n(\omega)a_{n+\tau}(\omega)^*]\rangle_n}{\sigma^2(\omega)}, \quad |\gamma_\tau(\omega)| \leq 1, \tag{3}$$

which is the time-averaged correlation coefficient. When $\gamma_\tau \approx 1$, errors align constructively; when $\gamma_\tau \approx 0$, they are uncorrelated. Regrouping by time-lag gives the exact identity:

$$\mathbb{E}[|e_n(\omega)|^2] = \sigma^2(\omega)\sum_{r=0}^{n-1}|\lambda(\omega)|^{2r} + 2\sigma^2(\omega)\sum_{\tau=1}^{n-1}\left(\sum_{r=0}^{n-1-\tau}|\lambda(\omega)|^{2r}\right)\Re[\lambda(\omega)^\tau\gamma_\tau(\omega)]. \tag{4}$$

When the model neither amplifies nor dampens frequencies significantly ($|\lambda(\omega)| \approx 1$), this simplifies to:

$$\mathbb{E}[|e_n(\omega)|^2] \approx n\sigma^2(\omega) + 2\sigma^2(\omega)\sum_{\tau=1}^{n-1}(n-\tau)\Re[\gamma_\tau(\omega)]. \tag{5}$$

Two regimes emerge: (i) Phase-locked injection ($\gamma_\tau \approx 1$) yields $O(n^2)$ growth—the harmonic spikes we observe. (ii) Decorrelated injection ($\gamma_\tau \approx 0$) yields only $O(n)$ growth.

With cycling, only frequencies in $\Omega_\cap$ see consistent alignment at every step. For a cycle $k \in \{4, 8, 16\}$, we have $\Omega_\cap = \Omega_{16}$ (the least common multiple lattice). On most frequencies $\omega \in \Omega_\cup \setminus \Omega_\cap$, the changing patch grid temporally thins and phase-misaligns injections, reducing $\Re[\lambda^\tau \gamma_\tau(\omega)]$ in (4). This increased decorrelation shifts a portion of the error growth from quadratic to linear, which might explain the spectral spike suppression.

In our linearized analysis, the expected error energy at a frequency omega after n steps decomposes into: (i) a term that is always linear in $n$, and (ii) a correlation term that depends on how phase-aligned error injections are across timesteps. For a fixed patch grid, on problematic harmonics the correlation term behaves like a sum $1+2+...+n$, i.e. $O(n^2)$, and dominates. With temporal cycling, the patch grid changes so that injections are typically not phase-aligned, driving the correlation term toward zero and leaving the linear term dominant. This is the sense in which cycling shifts part of the growth from quadratic to linear, consistent with the observed rollout error reduction. We emphasize that this analysis is approximate: the contribution is primarily empirical, and the theory is meant as a physically motivated explanation rather than a full proof.

We do not assume independence of $a_t$, claim exact rates for specific $\omega$, or assert optimality of our cycle. We chose powers of two in our patch sizes for computational efficiency and to avoid excess padding at image boundaries. While coprime sizes and a longer cycle might minimize $\Omega_\cap$ further (reducing the common harmonics), we leave exploration of such potential trade-offs to future work.

Intuitively, the key effect of alternating patch/stride sizes during rollout can be understood cleanly in the spatial domain. When a model is rolled out autoregressively with a fixed patch size, local errors tend to accumulate at the patch boundaries. Since the patch grid is fixed, the same locations see repeated errors step after step, causing these errors to reinforce and appear as grid-aligned patterns or "checkerboards" over long horizons. By contrast, when the patch size changes from one step to the next, the patch boundaries shift. This redistributes where those local errors are introduced: boundaries from one step do not align with those in the next step. As a result, error no longer accumulates coherently in the same locations and is instead spread more evenly across the spatial domain, thereby mitigating the structured buildup that causes grid artifacts.

Implementation-wise, the embedding dimension is fixed as in a standard ViT. Changing patch size only alters (1) the number of tokens and (2) the spatial footprint per token. During training we randomize over patch/stride sizes 4, 8, 16, so the network explicitly learns to interpret tokens from all these grids within a single shared embedding space. Small patches capture fine-scale structure; larger patches emphasize coarse/global structure. Alternating them over time lets the model repeatedly "refresh" fine and coarse scales, which empirically reduces checkerboard artifacts and improves long-horizon accuracy. So while a deterministic schedule may look "physically odd" at first glance, it is simply a structured way of varying the receptive field of an already multi-patch-trained model, not arbitrarily oscillating the physical representation.

Additionally, alternating patch sizes does not inject random noise, but it breaks the deterministic repetition of errors by varying the discretization pattern. Because the boundaries shift deterministically, the same spatial locations are no longer repeatedly exposed to boundary errors. This prevents errors from reinforcing in fixed positions and instead disperses them across the domain, reducing structured checkerboard artifacts over long rollouts. This leads to a more uniform error distribution (and reduces checkerboards) without requiring explicit noise.

## C  EXPERIMENT DETAILS

### C.1  DATASETS

The datasets that we benchmarked are taken from The Well collection Ohana et al. (2024). We selected 2D and 3D datasets with complex dynamics, ranging from biology (*Active Matter* Maddu et al. (2024)) to astrophysics (*Supernova Explosion* & *Turbulence Gravity Cooling* Hirashima et al. (2023a;b); *Turbulent Radiative Layer 2D* Fielding et al. (2020)) and fluid dynamics (*Rayleigh-Bénard* & *Shear Flow* Burns et al. (2020)) . Here is information about their resolution and physical fields (equivalent to a number of channels).

- *Active Matter*: Active matter systems are composed of agents, such as particles or macromolecules, that transform chemical energy into mechanical work, generating active forces

| Dataset | Resolution | # of fields/channels |
|---|---|---|
| *Active Matter* | $256 \times 256$ | 11 |
| *Rayleigh-Bénard* | $512 \times 128$ | 4 |
| *Shear Flow* | $128 \times 256$ | 4 |
| *Supernova Explosion* | $64 \times 64 \times 64$ | 6 |
| *Turbulence Gravity Cooling* | $64 \times 64 \times 64$ | 6 |
| *Turbulent Radiative Layer 2D* | $128 \times 384$ | 5 |

Table 6: Specifics of the datasets benchmarked.

or stresses. These forces are transmitted throughout the system via direct steric interactions, cross-linking proteins, or long-range hydrodynamic interactions, leading to complex spatiotemporal dynamics. These simulations specifically focus on active particles suspended in a viscous fluid leading to orientation-dependent viscosity with significant long-range hydrodynamic and steric interactions.

- *Rayleigh-Bénard*: Rayleigh-Bénard convection Rayleigh (1916); Wen et al. (2022) is a phenomenon in fluid dynamics encountered in geophysics (mantle convection Schubert et al. (2001), ocean circulation Siedler et al. (2001), atmospheric dynamics Holton & Hakim (2013)), in engineering (cooling systems Kakaç et al. (2012), material processing Poirier & Geiger (2016)), in astrophysics (interior of stars and planets Hansen et al. (2012)). It occurs in a horizontal layer of fluid heated from below and cooled from above. This temperature difference creates a density gradient that can lead to the formation of convection currents, where warmer, less dense fluid rises, and cooler, denser fluid sinks.

- *Shear Flow*: Shear flow phenomena Kundu et al. (2015); Wu & He (2021); Vinze & Michelin (2024) occurs when layers of fluid move parallel to each other at different velocities, creating a velocity gradient perpendicular to the flow direction. This can lead to various instabilities and turbulence, which are fundamental to many applications in engineering (e.g., aerodynamics Rizzi (2023)), geophysics (e.g., oceanography Smyth & Moum (2012)), and biomedicine (e.g. biomechanics Perinajová et al. (2021)). Code and software used to generate the data: https://github.com/RudyMorel/the-well-rbc-sf. We used this codebase to generate the dataset at a resolution of $1024 \times 2048$, and then downsampled that to $128 \times 256$ for memory purposes.

- *Supernova Explosion*: Supernova explosions happen at the end of the lives of some massive stars. These explosions release high energy into the interstellar medium (ISM) and create blastwaves. The blastwaves accumulate in the ISM and form dense, sharp shells, which quickly cool down and can be new star-forming regions. These small explosions have a significant impact on the entire galaxy's evolution.

- *Turbulent Radiative Layer 2D*: In astrophysical environments, cold dense gas clumps move through a surrounding hotter gas, mixing due to turbulence at their interface. This mixing creates an intermediate temperature phase that cools rapidly by radiative cooling, causing the mixed gas to join the cold phase as photons escape and energy is lost. Simulations and theories show that if cooling is faster (slower) than mixing, the cold clumps will grow (shrink) (Gronke & Oh, 2018; Abruzzo et al., 2024). These simulations Fielding et al. (2020) describe the competition between turbulent mixing and radiative cooling at a mixing layer.

- *Turbulence Gravity Cooling*: Within the interstellar medium (ISM), turbulence, star formation, supernova explosions, radiation, and other complex physics significantly impact galaxy evolution. This ISM is modeled by a turbulent fluid with gravity. These fluids make dense filaments, leading to the formation of new stars. The timescale and frequency of making new filaments vary with the mass and length of the system.

## C.2 MODEL CONFIGURATION

The core transformer processor consists of multiple stacked processing blocks, each composed of three key operations: (1) temporal self-attention, which captures dependencies across time steps, (2) spatial self-attention, which extracts spatial correlations within each frame, and (3) a multi-layer perceptron (MLP) for feature transformation. The number of such processing blocks varies across model scales, as detailed in Table 7. The full configuration is visualized in Figure 7.

Table 7: Model architecture configurations across different scales.

| Model | Embed Dim | MLP Dim | # Heads | # Blocks | Base Patch Size |
|-------|-----------|---------|---------|----------|-----------------|
| 7.6M  | 192       | 768     | 3       | 12       | [16, 16]        |
| 25M   | 384       | 1536    | 6       | 12       | [16, 16]        |
| 100M  | 768       | 3072    | 12      | 12       | [16, 16]        |

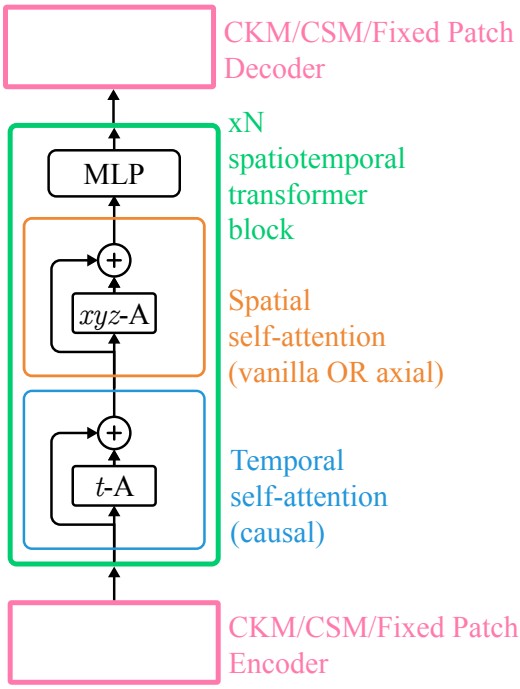

Figure 7: This visualization represents the overall architecture of our models. The CSM/CKM/Fixed patch are various strategies for the patch encoding and decoding. The patch encoder block is followed by $N$ transformer based spatio-temporal blocks containing sequential time attention, followed by full spatial attention, followed by MLP. The embedding dimension column in Table 7 represents the hidden dimension that the CSM/CKM/Fixed patch strategies embed the real space patches into.

## C.3 TRAINING CONFIGURATION - AXIAL VIT AND VANILLA VIT EXPERIMENTS

We use the following training configuration for the Axial ViT and Vanilla ViT experiments:

1. **Hardware and parallelism.** We train with a global batch size of 16 on 8 NVIDIA A100 80GB GPUs using PyTorch FSDP. No gradient accumulation.

2. **Optimizer and regularization.** We use Adam with learning rate $10^{-4}$ and weight decay $10^{-4}$, and set drop path to 0.1. We tuned the learning rate over $\{10^{-5}, 10^{-4}, 10^{-3}\}$.

3. **Training loss.** All models are trained with Normalized Mean Squared Error (NMSE), averaged over spatial dimensions and fields.

4. **Prediction target.** We use `delta` prediction, i.e., the model predicts the change in the field from the previous timestep. This affects training only; validation and test metrics are computed on reconstructed fields.

5. **Positional encoding.** We use RoPE (Su et al., 2021).

6. **Training details:** Each fixed-patch baseline ($p \in \{4, 8, 16\}$) is trained for 20,000 optimizer steps (200 epochs with epoch size of 100). The compute-elastic CSM/CKM model is trained

with a matched total number of optimizer updates equal to the sum of the three fixed-patch runs, sampling the stride/patch setting uniformly from $\{4, 8, 16\}$ at each update (same global batch size and hardware). Under uniform sampling, this results in approximately equal numbers of updates at each setting, matching the per-setting training budget of the fixed-patch suite.

7. **Data splits.** We use the official train/validation/test split provided by The Well dataset (Ohana et al., 2024). Accordingly, within each dataset, for each set of simulation parameters, we split initial conditions 80/10/10 across train/validation/test. For example, if there are 100 initial conditions for each of 5 parameter settings (each trajectory with 200 timesteps), we use 80 trajectories per parameter setting for training, 10 for validation, and 10 for testing.

## C.4 VRMSE METRIC

The variance scaled mean squared error (VMSE): it is the MSE normalized by the variance of the truth. It has been used for the benchmarking of the Well (Ohana et al., 2024).

$$\text{VMSE}(u, v) = \frac{\langle |u - v|^2 \rangle}{(\langle |u - \bar{u}|^2 \rangle + \epsilon)}.$$

We chose to report its square root variant, the VRMSE:

$$\text{VRMSE}(u, v) = \frac{\langle |u - v|^2 \rangle^{1/2}}{(\langle |u - \bar{u}|^2 \rangle + \epsilon)^{1/2}}.$$

Note that, since $\text{VRMSE}(u, \bar{u}) \approx 1$, having $\text{VRMSE} > 1$ indicates worse results than an accurate estimation of the spatial mean $\bar{u}$.

# D  BINNED SPECTRAL ERROR ANALYSIS

## D.1  DEFINITIONS

We assess robustness using the **binned spectral normalized mean squared error (BSNMSE)** (Ohana et al., 2024). BSNMSE measures the mean squared error between prediction and target after restricting both fields to a spatial frequency band $\mathcal{B}$, and then normalizing by the target energy in that band. Concretely, we first define the (unnormalized) **binned spectral mean squared error (BSMSE)** as

$$\text{BSMSE}_{\mathcal{B}}(u, v) = \left\langle |u_{\mathcal{B}} - v_{\mathcal{B}}|^2 \right\rangle, \tag{6}$$

where the band-limited field $u_{\mathcal{B}}$ is obtained by applying a bandpass filter in Fourier space,

$$u_{\mathcal{B}} = \mathcal{F}^{-1}\big[\mathcal{F}[u]\, \mathbf{1}_{\mathcal{B}}\big]. \tag{7}$$

Here, $\mathcal{F}$ denotes the discrete Fourier transform and $\mathbf{1}_{\mathcal{B}}$ is an indicator function selecting frequencies within $\mathcal{B}$.

For each dataset, we define three disjoint frequency bands $\mathcal{B}_1, \mathcal{B}_2$, and $\mathcal{B}_3$, corresponding to low, intermediate, and high spatial frequencies. We construct these bands by partitioning wavenumber magnitudes evenly on a logarithmic scale.

Finally, the **BSNMSE** is defined by normalizing the BSMSE by the target energy in the same band:

$$\text{BSNMSE}_{\mathcal{B}}(u, v) = \frac{\left\langle |u_{\mathcal{B}} - v_{\mathcal{B}}|^2 \right\rangle}{\left\langle |v_{\mathcal{B}}|^2 \right\rangle}. \tag{8}$$

A value of $\text{BSNMSE}_{\mathcal{B}} \geq 1$ indicates that the error at that scale is at least as large as the energy of the target signal, i.e., worse than predicting zero coefficients within band $\mathcal{B}$.

## D.2  BSNMSE FOR DIFFERENT DATASETS

Fig. 8 shows the next-step prediction BSNMSE of the test set for *Rayleigh-Bénard*, *Turbulent Radiative Layer 2D*, *Shear Flow* and *Supernova Explosion*. The spectrum is divided into three frequency bins: low, mid and high being the smallest, intermediate and largest frequency bins whose boundaries are evenly distributed in log space. Crucially, as also emphasized in the main text, we note that the fixed patch models (green) are *separately* trained models, while CKM (orange) and CSM (blue) (here shown is the flexified vanilla ViT model) are *single* models trained at randomized strides/patches.

Following our results in Tables 1 and 2 of the main text, our story is reflected in the BSNMSE metric as well. In other words, it is clear from the BSNMSE plot that CSM/CKM (coupled with vanilla ViT for this plot) enable the creation of a single flexible model while maintaining comparable accuracy to the fixed-patch baselines (and often improving it) across these cases. This supports training a single compute-elastic model that can be deployed at multiple inference-time tokenizations.

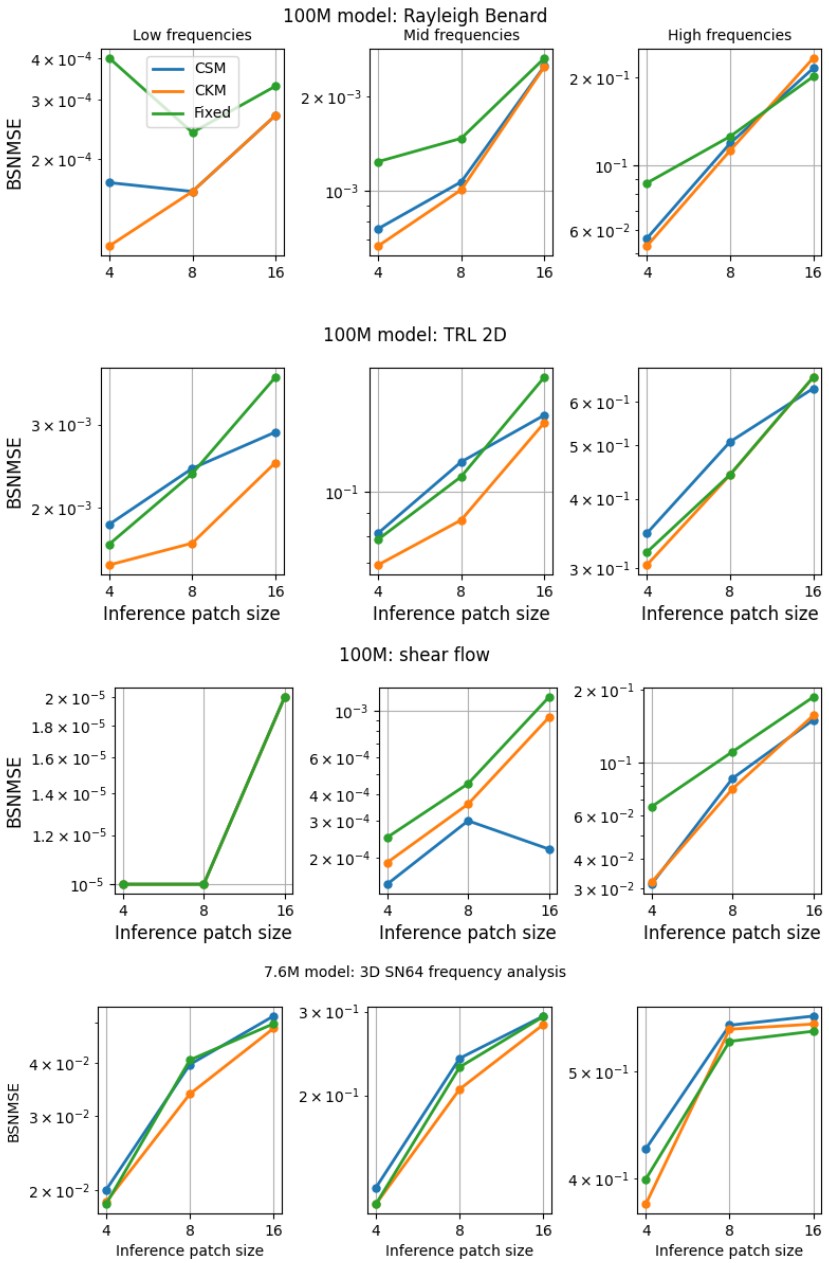

Figure 8: BSNMSE of the next-step prediction of the test set in the *Rayleigh-Bénard*, *Turbulent Radiative Layer 2D*, *Shear Flow* and *Supernova Explosion* datasets. The spectrum is divided into three frequency bins: low, mid and high being the smallest, intermediate and largest frequency bins whose boundaries are evenly distributed in log space. Crucially, as also emphasized in the main text, we note that the fixed patch models (green) are *separately* trained models, while CKM (orange) and CSM (blue) (here shown is the flexified vanilla ViT model) are *single* models trained at randomized stride/patch sizes.

Table 8: CViT next-step prediction performance across datasets and token counts, comparing CKM and Fixed patch size (f.p.s.). As before, the f.p.s. models are separately trained models while the flexible models are one single model capable of handling multiple patch resolutions at inference. Similar to the vanilla ViT and axial ViT architectures, we find that CViT consistently gains from smaller patch sizes, can be flexified for practical deployment and an improvement in accuracy.

| Dataset | Tokens/Patch | Flexible CViT | Fixed CViT |
|---|---|---|---|
| *Shear Flow* | 2048/4 | **0.120** | 0.184 |
| | 128/16 | **0.127** | 0.319 |
| *Turbulent Radiative Layer 2D* | 3072/4 | 0.302 | 0.299 |
| | 192/16 | **0.340** | 0.364 |
| *Active Matter* | 4096/4 | **0.054** | 0.110 |
| | 256/16 | **0.065** | 0.127 |

## E CViT EXPERIMENTS

**CViT experiments.** We further evaluate our flexible patching strategy within the Continuous Vision Transformer (CViT) architecture (Wang et al., 2025b), a recent vision transformer-based model tailored for PDE surrogate modeling. CViT features a patch-based transformer encoder coupled with a novel, grid-based continuous decoder that directly queries spatiotemporal locations. As the decoder operates without patch-based tokenization, we apply our kernel modulation (CKM) only to the encoder, leaving the rest of the architecture unchanged.

A flexified CViT therefore refers to the encoder being a CKM. For CViT (Wang et al., 2025b), whose original design features a single-stage encoder (instead of multi-stage hMLP we used for our vanilla ViT and axial ViT experiments) and a decoder based on grid-based querying, CKM is applied only to the encoder, for the single stage; because the idea is to see if flexification on top of the base fixed patch architecture helps or not. These use cases show CKM's modularity and adaptability: it can flexify arbitrary convolutional patching pipelines, without requiring architectural changes or any overengineering to the base convolutional or attention mechanism or task head.

Since CViT is designed for 2D problems, we restrict our evaluation to 2D datasets, ensuring a fair and focused test of patch flexibility without altering any other model components. This experiment shows that our flexible tokenization strategy is architecture-agnostic and can be modularly integrated into advanced PDE models like CViT.

Table 8 shows that a flexified CViT consistently outperforms fixed patch size CViT models across four diverse 2D PDE datasets. As in earlier experiments, reducing patch size (e.g., from 16 to 4) increases token count and compute, but yields significantly improved accuracy. Importantly, a flexified CViT provides this accuracy-compute trade-off at inference without retraining, demonstrating the modularity and architectural agnosticism of Overtone. This result also shows how adaptive patching can be extended to complex hybrid architectures like CViT, unlocking new modes of inference-time flexibility in scientific surrogate modeling.

In terms of rollouts, since CViT uses a query based continuous decoder, instead of a patch based decoder, it does not inherently suffer from the issues of patch artifacts. Even for that case, Table 9 shows that flexified CViT performs a lot better than the corresponding static patch version. This again shows that our flexification approach is useful even for cases of hybrid encoder-decoder architectures where only the encoder is patch based, but the decoder is not. This reinforces the architecture agnostic-ness and the SOTA performance of our flexible models.

**Model details for CViT.** For the base CViT architecture, we utilized the CViT-S configuration in (Wang et al., 2025b) (See Table 1 of this paper), setting: Encoder layers = 5, Embedding dim = 384, MLP width = 384, Heads = 6. We set the embedding grid size for CViT to be the same as the dataset size.

Table 9: 10-step rollout VRMSE comparison across datasets for CViT vs. fixed patch size 16 (f.p.s) models. $\Delta$ indicates the best percentage improvement (i.e., lowest VRMSE vs. fixed patch baseline).

| Dataset | Flexible CViT | Fixed p.s. 16 | $\Delta$ |
|---|---|---|---|
| *Shear Flow* | **0.063** | 0.2598 | +75.7% |
| *Turbulent Radiative Layer 2D* | **0.540** | 0.568 | +4.9% |
| *Active Matter* | **0.78** | 3.33 | +76.57% |

**CViT Patch Embeddings.** The ViT encoder takes as input a gridded representation of the input function $u$, yielding a spatio-temporal data tensor $\mathbf{u} \in \mathrm{R}^{T \times H \times W \times D}$ with $D$ channels. The model patchifies the input into tokens $\mathbf{u}_p \in \mathrm{R}^{T \times \frac{H}{P} \times \frac{W}{P} \times C}$ by tokenizing each 2D spatial frame independently, following the process used in standard Vision Transformers (Dosovitskiy et al., 2020). The patching process involves a single convolutional downsampling layer. We add trainable 1D temporal and 2D spatial positional embeddings to each token. These are absolute position encodings, unlike the RoPE encodings we used for vanilla and axial ViT experiments.

$$\mathbf{u}_{pe} = \mathbf{u}_p + \mathrm{PE}_t + \mathrm{PE}_s, \quad \mathrm{PE}_t \in \mathrm{R}^{T \times 1 \times 1 \times C}, \quad \mathrm{PE}_s \in \mathrm{R}^{1 \times \frac{H}{P} \times \frac{W}{P} \times C}. \quad (9)$$

The original paper (Wang et al., 2025b) has already detailed the above description of CViT. Our main experiments were to flexify this CViT architecture and show that under a given compute budget, training a flexible CViT is much more advantageous than training multiple fixed patch CViT models. To this end, we adapt CKM to flexify the convolutional layer of CViT, randomizing the patch size at the encoding stage. Additionally, since base CViT has absolute learned position embedding, we resized the position embedding at each step of the forward pass as well using bilinear interpolation to make it compatible with the randomized patch size.

**Goals with flexified CViT.** Our goal with the CViT experiments is to show that, similar to the axial ViT and vanilla ViT results, CViT also benefits from flexification. In other words, with a fixed compute budget, it is much more beneficial to train a flexible CViT model than to train and maintain multiple static resolution CViT models. This leads to the creation of a flexible CViT, similar to the flexible versions of axial and vanilla ViTs we created earlier.

First we show that similar to axial and vanilla ViT, CViT also consistently gains in accuracy as the patch size, aka token counts is decreased (Table 8). Additionally, we show that our alternating patch strategy for CViT leads to dramatically better accuracies than the fixed patch models (Table 9). Through these experiments, we achieve the flexification of this recent architecture. This experiment further shows the architecture agnostic-ness of Overtone's methods; meaning we can make flexification compatible with a range of base architectures, as exemplified by our results in the main text, augmented the flexification of this advanced hybrid architecture like CViT. Flexified CViT is compute-adaptive meaning it can adapt to various downstream compute/accuracy requirements at inference after being trained only *once*, eliminating the need to train and maintain multiple models.

**Training details for CViT experiments.** We follow a consistent training protocol to ensure a fair comparison between fixed-patch and flexified CViT models. All models use the CViT-S architecture from (Wang et al., 2025b), configured with 5 encoder layers, an embedding dimension of 384, MLP width 384, and 6 attention heads. We match the spatial embedding grid size to the input resolution of each dataset.

We fix the learning rate at $10^{-4}$, chosen after a hyperparameter search over $\{10^{-2}, 10^{-3}, 10^{-4}\}$. We set weight decay to $10^{-5}$. We choose batch size to fully utilize GPU memory for each dataset. We conduct training on a single NVIDIA A100 80GB GPU.

Due to compute limitations, training epochs vary across datasets, but remain consistent within each dataset. Concretely:

- For *Shear Flow*, we train each fixed-patch model (patch sizes 4, 8, and 16) for 30 epochs.
- For *Active Matter*, we train each for 40 epochs.

- For *Turbulent Radiative Layer 2D*, we train each for 160 epochs.

We train the corresponding flexified CViT models (i.e., with CKM) with the same total compute budget as all three fixed-patch models combined. This allows a direct, compute-equivalent comparison. As described in Section 3.1, this setup addresses a practical question: under a fixed training budget, is it better to train multiple static models or one flexible model? If the flexible model enables training without any loss in validation accuracy, we have a flexible model by training only once eliminating the need to train and maintain multiple fixed patch models.

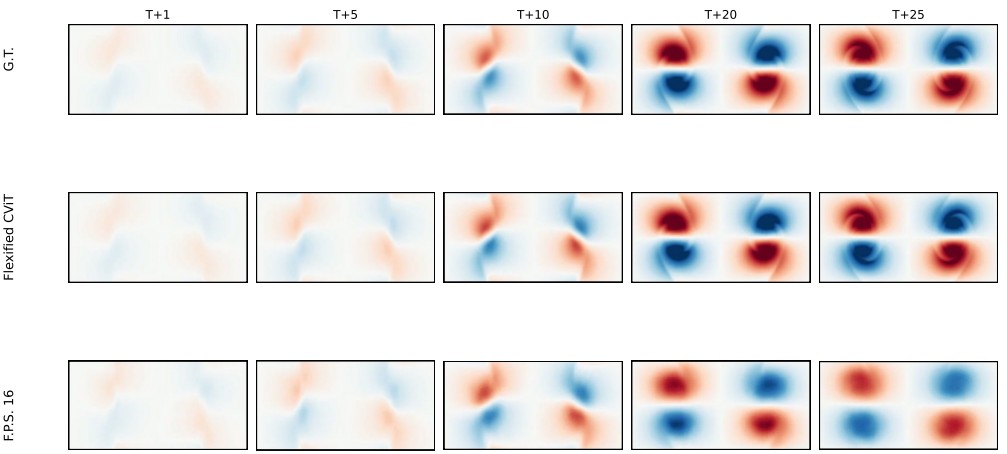

Figure 9: Rollout for the $v_y$ *Shear Flow* dataset for the CViT model. Top: Ground Truth (G.T.); Middle: Flexified CViT; Bottom: Fixed Patch 16 (F.P.S. 16) models.

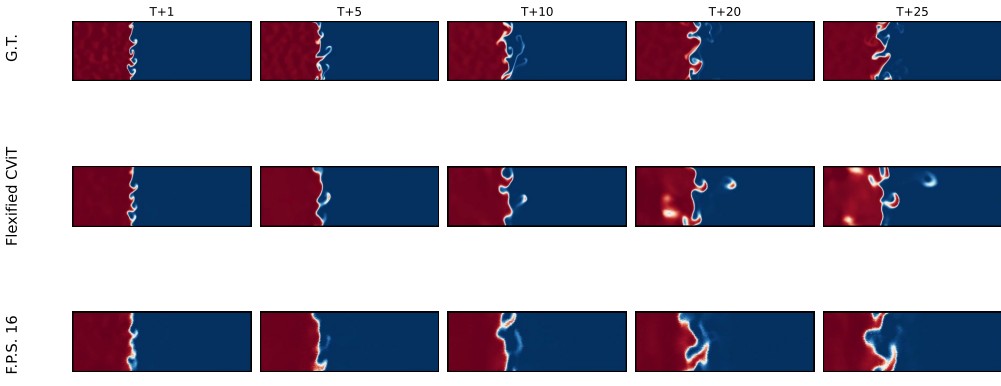

Figure 10: Rollout for the density *Turbulent Radiative Layer 2D* dataset for the CViT model. Top: Ground Truth (G.T.); Middle: Flexified CViT; Bottom: Fixed Patch 16 (F.P.S. 16) models.

## F  PATCH ARTIFACTS

Patch artifacts are widely present in patch based PDE surrogates. In this section, we will provide extended evidence for the existence of these artifacts. Note that these patch artifacts are a problem for the architectures where both the encoder and decoder are patch based, so this is not a problem for an architecture like CViT.

### F.1  OUR EXPERIMENTS

We have investigated and found these artifacts to present in a range of patch based ViT architectures. Below we will show patch artifacts arising in three different types of ViT based architectures, in different kinds of datasets: axial ViT (Ho et al., 2019), full ViT (Dosovitskiy et al., 2020) and Swin (Liu et al., 2021).

Figures 11 and 12 reflect the presence of these artifacts in rollouts of two example fields of the *Active Matter* and *Turbulence Gravity Cooling* datasets respectively. Figure 13 also shows artifacts arising when the base attention is changed to swin. Note that having a shifted window attention does not change the fact that the encoder and decoder still patchify with fixed size patches in conventional ViTs. Therefore even though attention can happen in a shifted-window format this fundamental limination imposed by the encoding-decoding process still remains.

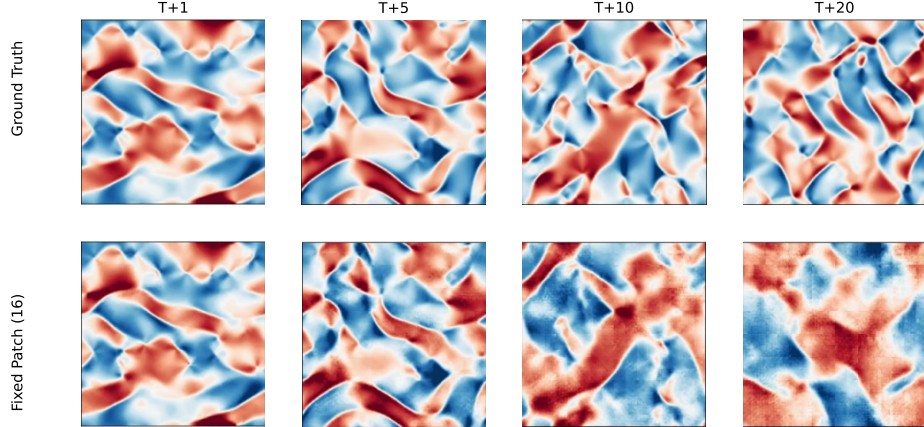

Figure 11: Artifacts arising in fixed patch models of the axial+ViT fixed patch 16 model. Results are shown for the $E_{xx}$ field of *Active Matter* dataset.

We have additionally tested that these artifacts are a core problem with the standard patch based ViTs, and they do not simply arise due to training details, such as different learning rates. Figure 14 shows that patch artifacts are agnostic to learning rates. This is expected because they arise because of the inherent limitation of fixed patch grids.

We have additionally tested that these artifacts also arise in neural operator models like AFNO (Guibas et al., 2021), which also has a patch based encoder and a decoder. We took the AFNO model provided as a baseline comparison in the Well benchmark suite (Ohana et al., 2024), and trained it on the *Turbulent Radiative Layer 2D* dataset, and as expected due to its patch based encoding and decoding, patch artifacts arise here as well as shown in Figure 15.

### F.2  OTHER PAPERS

These artifacts have also been seen in models where patch based encoders and decoders are used. For example, a recent study on pretraining on the dataset of the Well by Morel et al. (Morel et al., 2025). The artifacts can be seen in Figures 10 (3rd and 4th row), Figure 11 (3rd and 7th row), Figure 12 (3rd

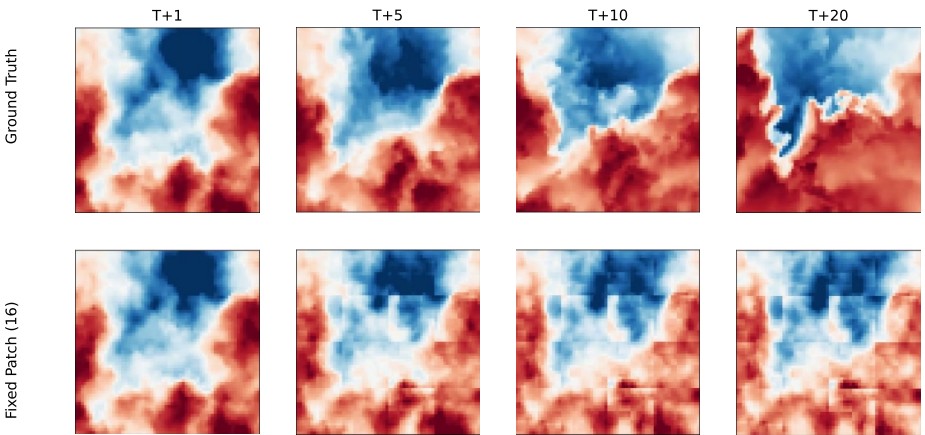

Figure 12: Artifacts arising in fixed patch models of the vanilla+ViT fixed patch 16 model. Results are shown for the $v_x$ field of *Turbulence Gravity Cooling* dataset.

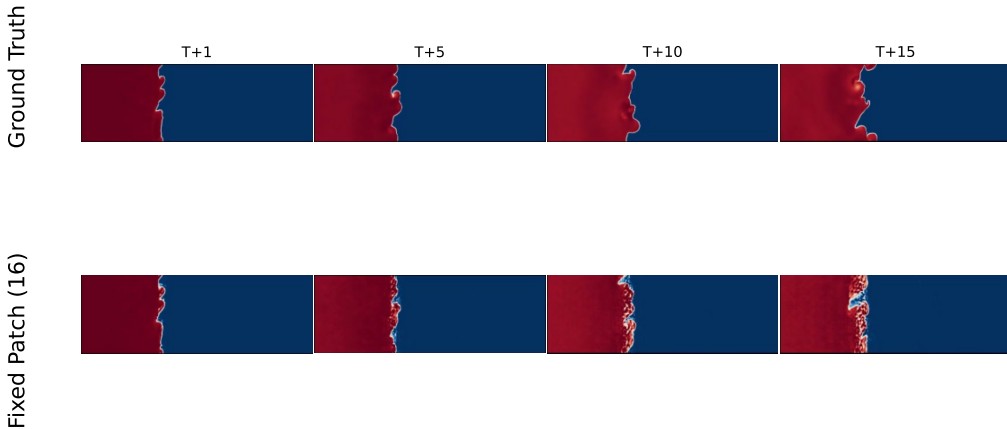

Figure 13: Artifacts arising in a swin transformer model where the spatial module is replaced by a swin transformer, keeping everything else the same. Results are shown for the density field of *Turbulent Radiative Layer 2D* dataset.

and 7th row)and Figure 13 (3rd, 4th and 7th row). In this paper, these artifacts are most common in the Multiple Physics Pretraining, MPP (McCabe et al., 2023) approach, which has both a patch based and a decoder.

The broad goal in this section is to show that these patch based artifacts are quite commonly seen in models that incorporate patch based encoding and decoding. Researchers can incorporate our flexible patching/striding methods into a wide variety of these models, and we anticipate that the rollout strategies we discussed in the main text can be improve the overall stability, accuracy and mitigate patch artifacts in these models. Due to limited compute, experiments with every possible architecture is difficult, but our study opens up possibilities in exploring this direction.

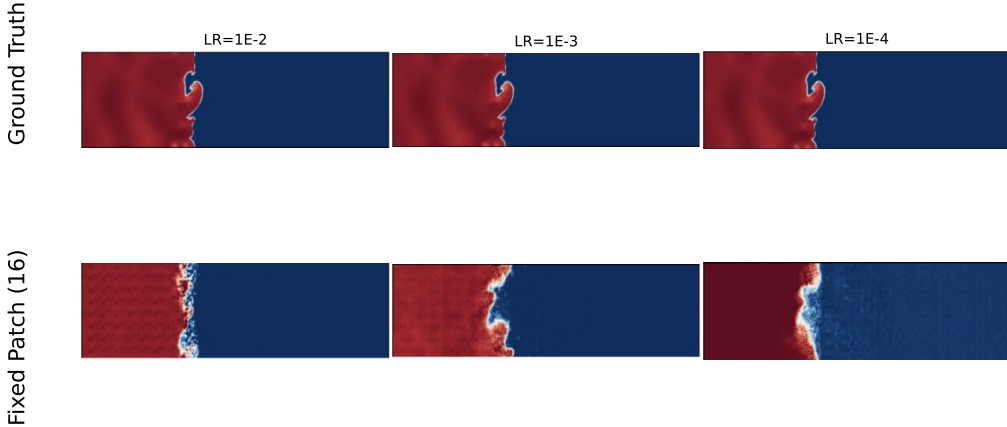

Figure 14: Artifacts in the density field of *Turbulent Radiative Layer 2D* dataset at rollout step 20. Artifacts appear regardless of the learning rate used for training.

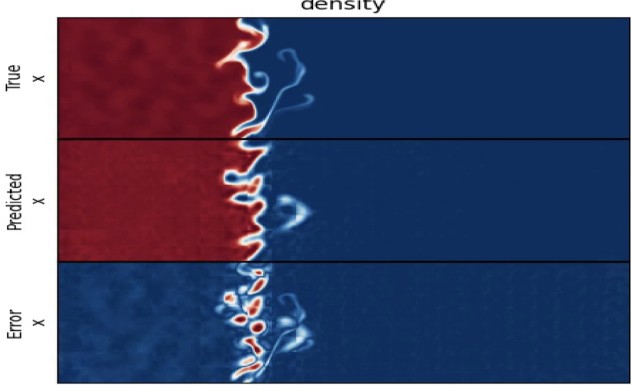

Figure 15: Artifacts in the density field of *Turbulent Radiative Layer 2D* dataset at rollout step 20. This is the AFNO (Guibas et al., 2021) model. Patch artifacts also arise here because even though its a neural operator based method, but the encoder and decoder still uses ViT based patch encoding/decoding.

## G    BASELINES USED AND MORE ROLLOUTS

Section 3.4 (Table 4) provided baseline comparisons with respect to a range of SOTA models which are not patch based. On the neural operator side, we included FFNO (Tran et al., 2023). We also test against a SOTA convolution based model, SineNet (Zhang et al., 2024b). Additionally, we tested against a recent Physics Attention based scheme, Transolver (Wu et al., 2024). In addition to these comparisons, we also perform comparisons against baseline models provided with the Well dataset module, and the well paper (Ohana et al., 2024). These models were TFNO (Kossaifi et al., 2023), FNO (Li et al., 2021), U-Net (Ronneberger et al., 2015) and CNext-U-Net (Liu et al., 2022). We reported the best baseline results out of these comparisons against the Well benchmarks. These non-patch baselines covered diverse comparisons. Below we describe these models and the settings used to make comparisons:

**SineNet:**    This model consists of multiple sequentially connected U-shaped network blocks called *waves* (Zhang et al., 2024b). While traditional U-Net architectures use skip connections to support multi-scale feature processing, the authors argue that forcing features to evolve across layers leads to temporal misalignment in these skip connections, ultimately limiting model performance. In contrast, SineNet progressively evolves high-resolution features across multiple stages, reducing feature misalignment within each stage and improving temporal coherence. This model has achieved SOTA performance on a range of complex PDE tasks. Refer to the original paper for more details.

We used their open source code from: `https://github.com/divelab/AIRS/blob/main/OpenPDE/SineNet/pdearena/pdearena/modules/sinenet_dual.py`. We use the SineNet-8-dual model with configurations outlined in their codebase, taken from `https://github.com/divelab/AIRS/blob/main/OpenPDE/SineNet/pdearena/pdearena/models/registry.py` (sinenet8-dual model). The network has 64 hidden channels with 8 waves. We set the learning rate to $2 \times 10^{-4}$, set after doing a coarse tuning among $10^{-4}, 2 \times 10^{-4}, 5 \times 10^{-4}, 10^{-3}$. The resulting model has around 35M parameters. We performed training on a single Nvidia A100-80GB GPU. We set batch size at 16.

**F-FNO:**    Fourier neural operators (Li et al., 2021) perform convolutions in the frequency domain and were originally developed for PDE modeling. We compare to a state-of-the-art variant, the factorized FNO (Tran et al., 2023) which improves over the original FNO by learning features in the Fourier space in each dimension independently, a process called Fourier factorization. We used the open source FFNO codebase provided in the original publication. We used a small version of the FFNO model with 4 layers, 32 fourier modes and 96 channels. This amounted to a model size of $\sim$ 7M. During training, a learning rate of $10^{-3}$ is utilized after a parameter tuning in the range $10^{-4}$, $10^{-3}$ and $10^{-2}$. Training was performed on a single Nvidia A100-80GB GPU. We set batch size to 16.

**Transolver:**    In this model (Wu et al., 2024), the authors propose a new Physics-Attention scheme that adaptively splits the discretized domain into a series of learnable slices of flexible shapes, where mesh points under similar physical states is ascribed to the same slice. The model achieved SOTA performance in a range of benchmarks including large-scale industrial simulations, including car and airfoil designs. We use their open source codebase and adapt it to train on the Well. We used 8 hidden layers, 8 heads, 256 hidden channels, and 128 slices. This amounted to a model of size 11M. This configuration matches the configuration presented in their main experiments. Training was performed on a single Nvidia A100-80GB GPU. We set batch size to 16.

As shown in Table 4, transolver generally struggled with the PDE tasks we tested on. We tested by tuning the learning rates varying between $10^{-3}, 10^{-4}$ and $10^{-2}$ without any significant improvement in performance. In the original paper, for autoregressive prediction tasks, they tested on $64 \times 64$ Navier Stokes, and downsampled their Darcy data into $85 \times 85$ resolution. Our datasets, on the other hand all exceed $128 \times 128$, and therefore the training was much slower. In order to have a fair comparison, we do not downsample our datasets for training the transolver model. The transolver code we used, is taken directly from their codebase.

**TFNO, FNO, U-Net, CNext-U-Net**    : The Well benchmarking suite provides these additional baselines with default configs together (Ohana et al., 2024). We use these default baselines. Neural-

operator v0.3.0 provides TFNO. Defaults are 16 modes, 4 blocks and 128 hidden size. U-net classic has default configs of: initial dimension = 48, spatial filter size = 3, block per stage = 1, up/down blocks = 4, bottleneck blocks = 1. CNext-U-Net has configs: spatial filter size = 7, initial dimension = 42, block per stage = 2, up/down blocks = 4 and bottleneck blocks = 1. Finally, the FNO model has 16 modes, 4 blocks and 128 hidden size. As noted before, the Well baseline package provides these model configs and more details can be found there. We showed the best benchmark results from the Well baseline suite and found that Overtone's models consistently outperformed these baselines. Table 10 shows the additional baselines.

Table 10: Next-step VRMSE across Well benchmark datasets using baselines provided in the Well.

| Dataset | FNO | TFNO | U-Net | CNext-U-Net |
|---|---|---|---|---|
| *Active Matter* | 0.262 | 0.257 | 0.103 | 0.033 |
| *Rayleigh-Bénard* | 0.355 | 0.300 | 0.469 | 0.224 |
| *Shear Flow* | 0.104 | 0.110 | 0.259 | 0.105 |
| *Supernova Explosion* | 0.378 | 0.379 | 0.306 | 0.318 |
| *Turbulence Gravity Cooling* | 0.243 | 0.267 | 0.675 | 0.209 |
| *Turbulent Radiative Layer 2D* | 0.500 | 0.501 | 0.241 | 0.226 |

We trained all of the above mentioned baselines such that they see the same number of observed samples as our flexible models (see Appendix C.3), ensuring that the comparison is fair.

Out of all of these baselines, we found that F-FNO and SineNet gave the most competitive results to Overtone's models, as shown in Table 4 on two of our complex 2D datasets. Given the competitive performance of F-FNO and SineNet on the two datasets shown in the text, we performed additional comparisons of these baselines on more datasets shown in Table 11. Clearly, these additional comparisons tell the same story that out flexible architecture-agnostic models outperform SOTA architectures on complex physics tasks.

| | FFNO | SineNet | Vanilla ViT (Ours) | | Axial ViT (Ours) | |
|---|---|---|---|---|---|---|
| Dataset | 7M | 35M | CSM (7M) | CKM (7M) | CSM (45M) | CKM (45M) |
| *Rayleigh-Bénard* | 0.94 | > 1 | **0.137** | **0.192** | **0.1664** | **0.215** |
| *Active Matter* | 0.567 | 0.760 | **0.563** | **0.522** | **0.384** | **0.390** |

Table 11: 10-step rollout VRMSE comparison across patch-free models and our patch-based surrogate variants. We report scores for both 7M and 100M parameter versions of CSM and CKM. Results are shown for vanilla and axial self-attention scheme.

Figure 16 visually shows that CSM/CKM + axial ViT models (45M size) are significantly better visually than the SineNet 35M model in long rollout predictions as well. These rollouts for a highly complex and chaotic *Rayleigh-Bénard* dataset. Our flexible models achieve state-of-the-art performance here.

To prove the effectiveness of Overtone's flexible models further, we show the 44th step rollout of all four fields of the highly complex and chaotic *Rayleigh-Bénard* dataset in Figure 17 for CSM + vanilla ViT, Figure 18 for CKM + vanilla ViT and Figure 19 for FFNO. We chose to show FFNO because it performed best among neural operator models in our baseline suite. All of the models have a comparable size of about 7M paramaters. Along with the metrics noted in Table 11, we find compelling visual evidence that CSM/CKM perform dramatically better than a competitive model like F-FNO at a similar parameter level. This demonstrates Overtone's strong performance.

**More rollouts.** Additionally, provide some more rollouts for different datasets for our flexible models. Figures 20 and 21. Consistent with the message of our paper, we find that the CSM and CKM strategies help to stabilize and significantly improve rollouts compared to fixed patch models; along with beating a range of state-of-the art models that we have extensively discussed.

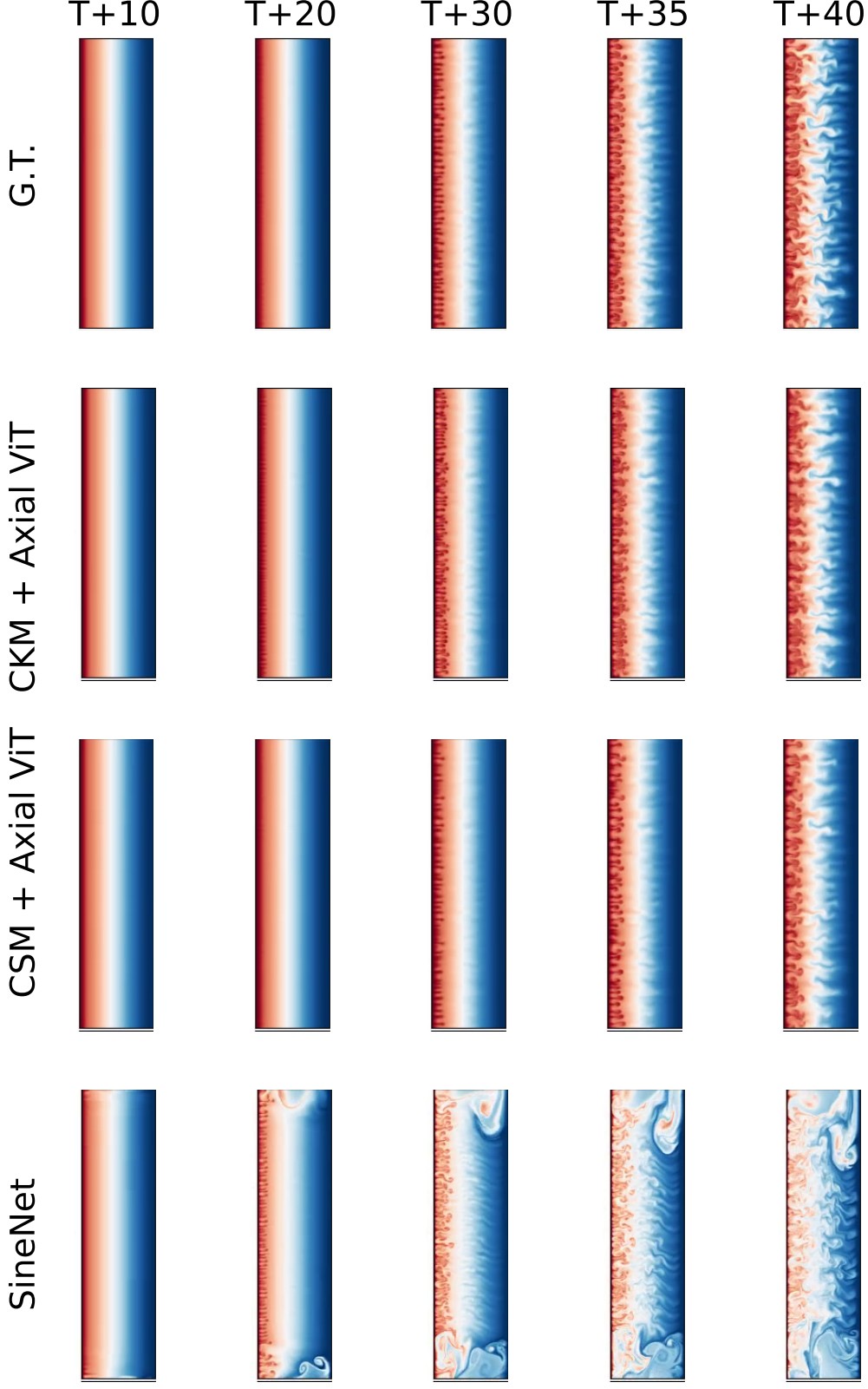

Figure 16: Figure showing a rollout for the *Rayleigh-Bénard* dataset. From top to bottom: ground truth (G.T.), CSM + axial ViT, CKM + axial ViT, SineNet. They all have comparable sizes in the order of 40M parameters.

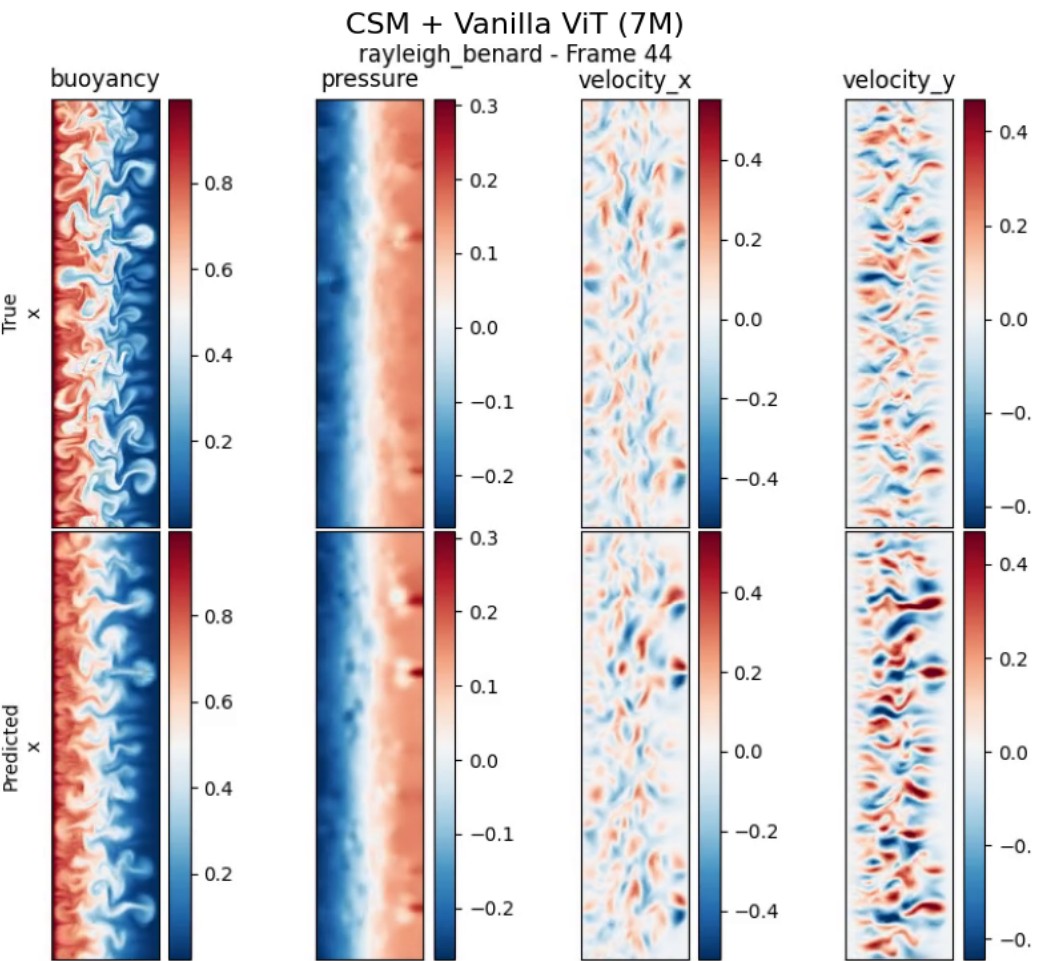

Figure 17: Figure showing the 44th step rollout of the 7M CSM + Vanilla ViT model.

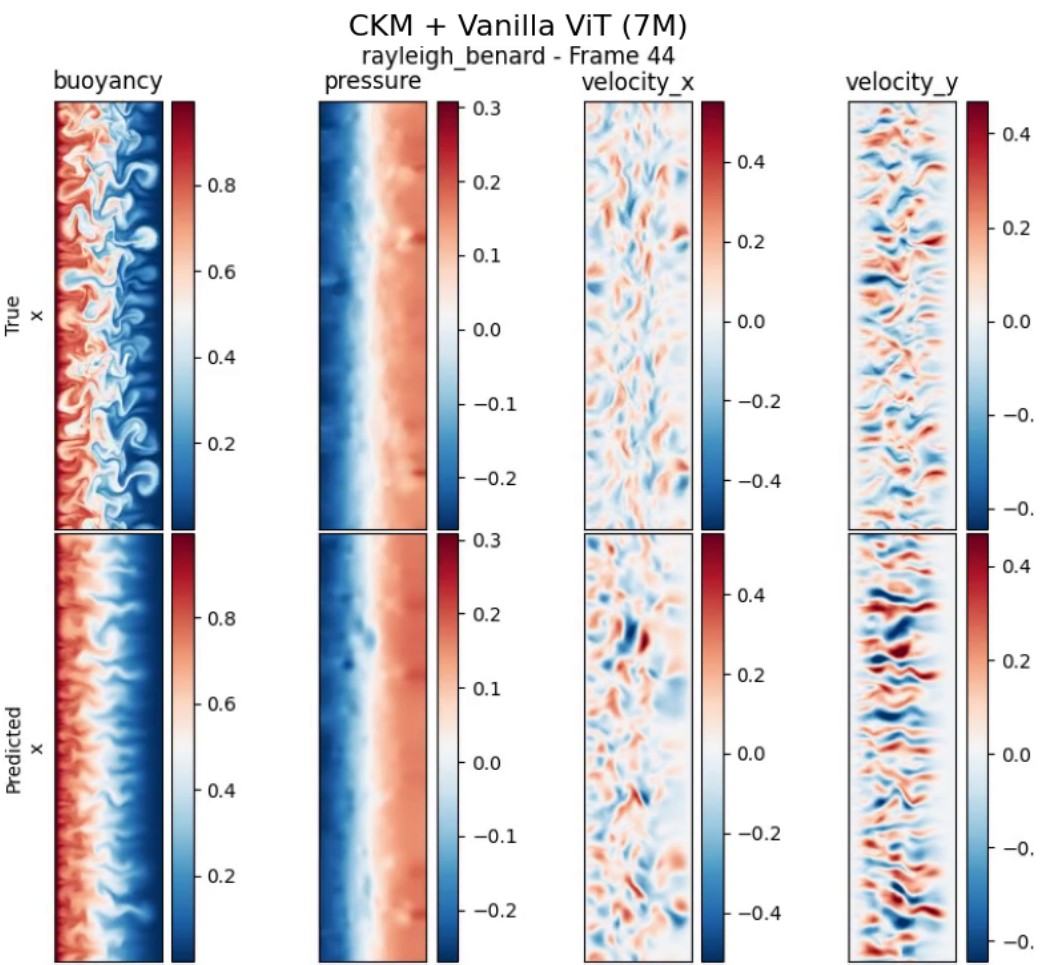

Figure 18: Figure showing the 44th step rollout of the 7M CKM + Vanilla ViT model.

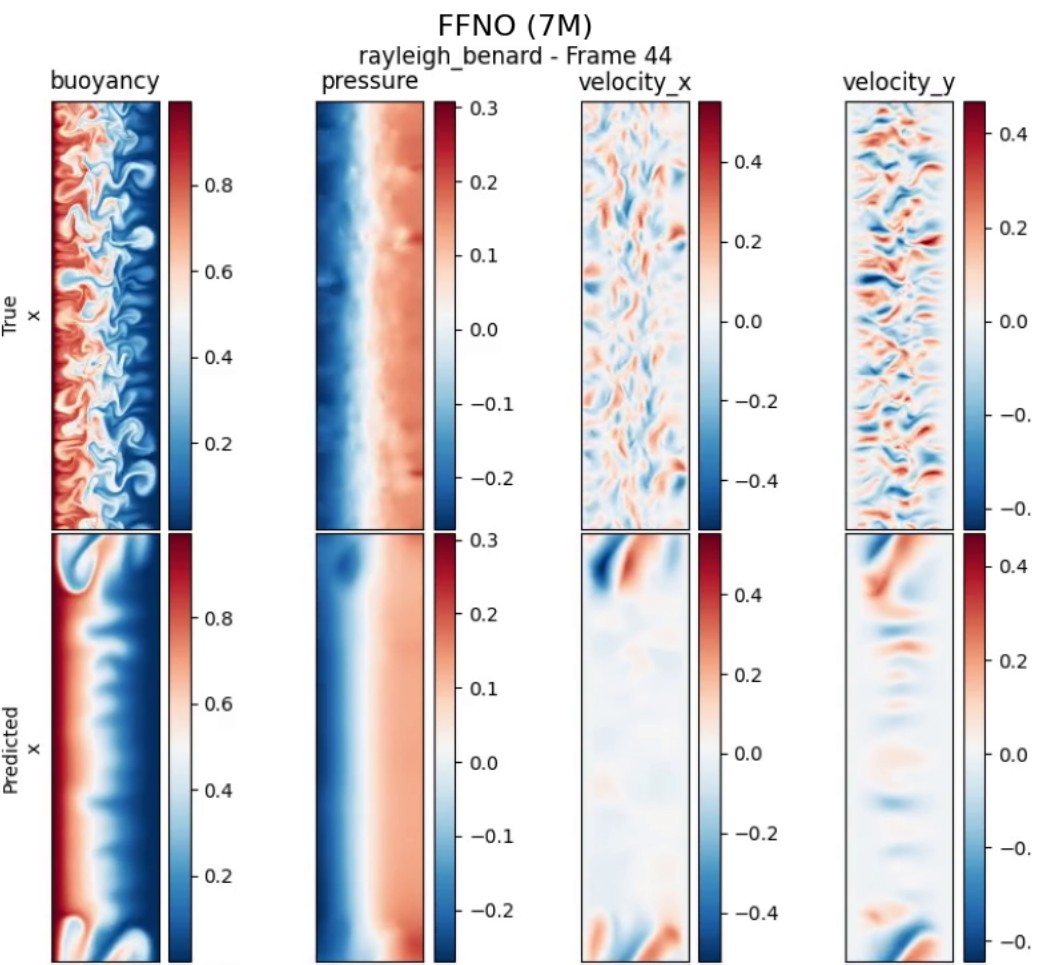

Figure 19: Figure showing the 44th step rollout of the 7M F-FNO model.

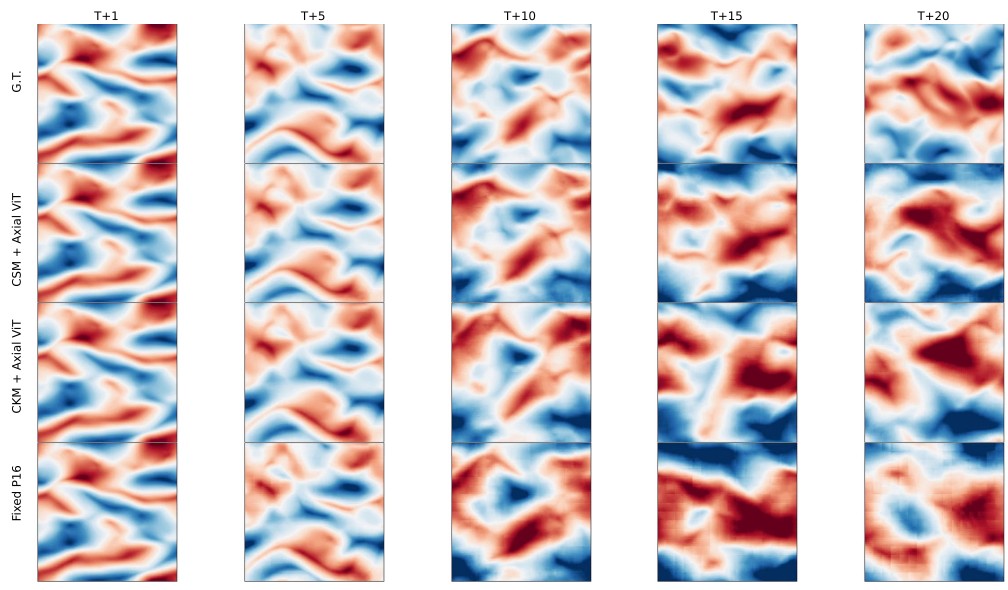

Figure 20: Rollouts of the *Active Matter* dataset.

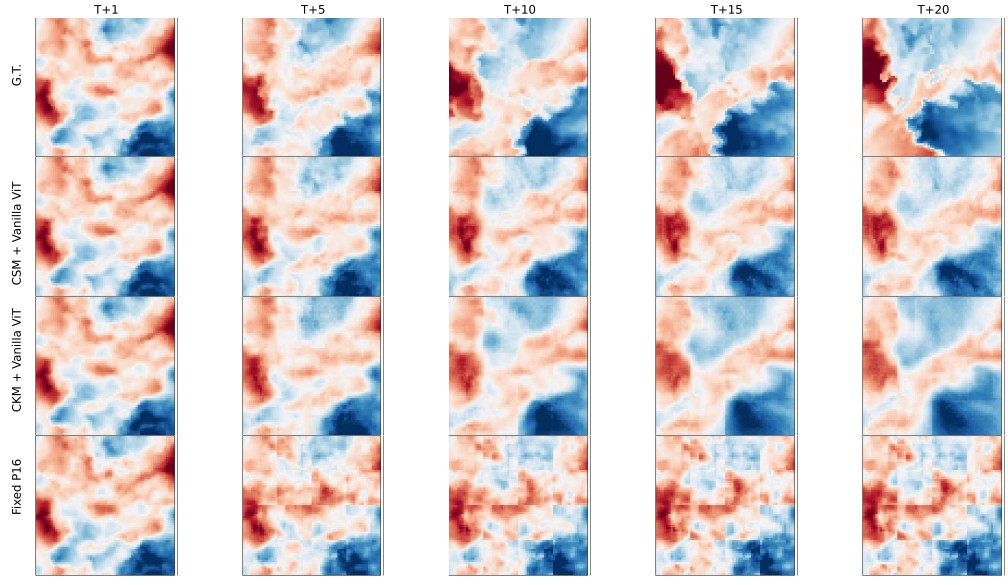

Figure 21: Rollouts of the *Turbulence Gravity Cooling* dataset.

## H  PHYSICS INFORMED BEHAVIOR

We investigate whether the models respect physical constraints such as mass and momentum conservation. First, we show a long (100 steps) trajectory rollout for shear flow for 100 steps in Fig. 22. We find that in accordance with the main text, fixed patch 16 model develops artifacts already by step 40, while CSM/CKM has much greater long horizon stability.

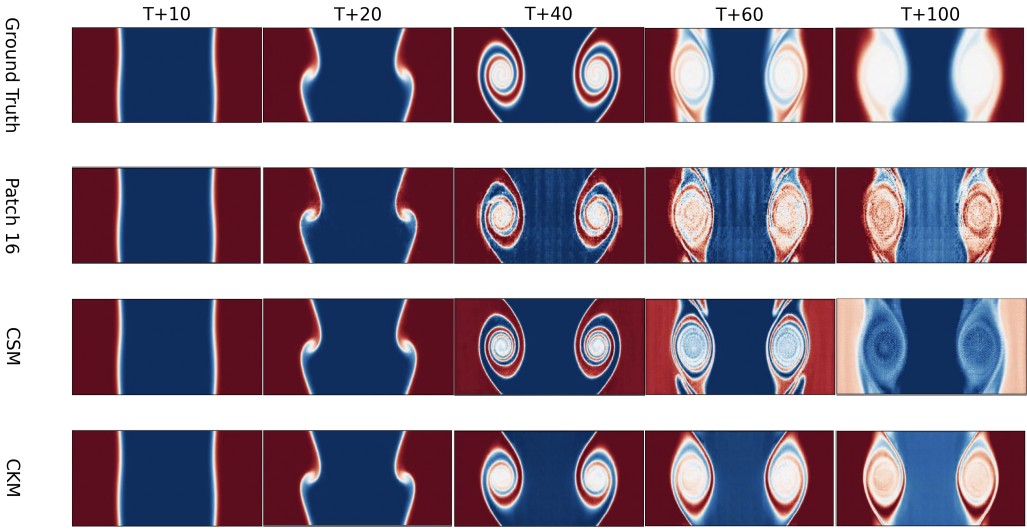

Figure 22: Long horizon rollout for shear flow

To probe physics based constraints, we performed new diagnostics on our CSM and CKM models on the incompressible shear flow dataset and measured the mass conservation,

$$|\nabla \cdot \mathbf{u}|,$$

and momentum conservation,

$$|\partial_t \mathbf{u} - \nu \Delta \mathbf{u} + \mathbf{u} \cdot \nabla \mathbf{u} + \nabla p|,$$

over 50 autoregressive rollout steps. Averaged over four randomly chosen test set trajectories at Reynolds number

$$\mathrm{Re} = 5 \times 10^4$$

(provided in the Well test set), the normalized deviations of these quantities relative to the ground truth numerical simulations are approximately 0.5% (CSM) and 0.75% (CKM) for momentum, and 7.6% (CSM) and 6.7% (CKM) for mass.

These results show that momentum conservation is tracked very closely, while mass conservation errors are somewhat larger (as is typical for autoregressive surrogates) but remain bounded ($< 10\%$ error) over long rollouts, which is impressive for long rollouts in emulators.

These results indicate that our models remain stable and preserve conserved quantities (at levels acceptable for measuring accuracy in fields like astrophysics and fluid dynamics) well over long horizons.

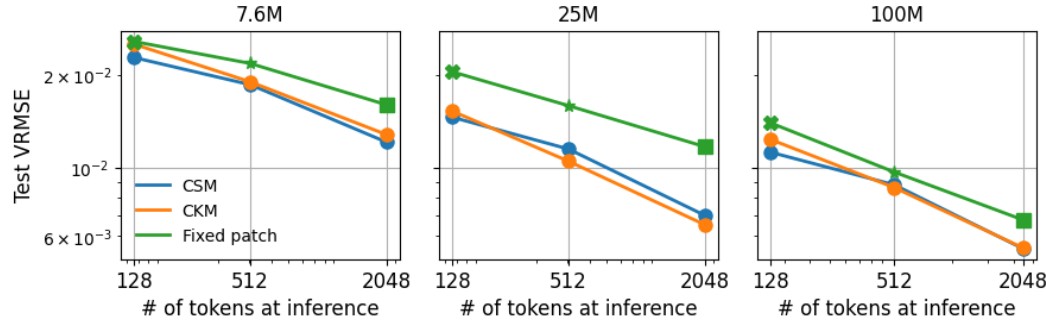

Figure 23: Comparison of next-step test VRMSE as a function of the number of tokens at inference for convolutional stride modulation models (CSM), convolutional kernel modulation models (CKM), and individually trained fixed patch models (7.6M, 25M and 100M parameters) for the *Shear Flow* dataset.

## I ABLATION STUDIES

### I.1 MODEL SIZE ABLATION

For the *Shear Flow* dataset, we trained the CSM/CKM and fixed patch (for the vanilla ViT self-attention) models. The next-step test set VRMSE are shown in Figure 23. The results are to be interpreted in the same way as Figure 3; i.e. the green points are separately trained static patch models that we evaluate at the corresponding training patch/size or token counts at inference. CSM and CKM on the other hand are a *single* model that we evaluate at different token counts at inference. The plots show that CSM/CKM outperform the corresponding static patch counterparts consistently at a range of parameter counts varying from 7M to 100M. This ablation shows that our story is consistent across scaling parameter count.

For further investigation, we repeated the ablation experiment for the smallest 7M model for our other benchmark datasets, shown in Table 12. We find CSM/CKM to be consistently performing competitively with their fixed patch counterparts across our benchmark datasets. This shows, again that flexification is valuable to be incorporated into current patch based PDE surrogates, since it provides a flexible choice of patching/striding parameter allowing tunability with various downstream tasks and compute adaptivity in PDE surrogates.

| Dataset | Patch size | Test VRMSE 7M Vanilla ViT | | |
|---|---|---|---|---|
| | | CSM | CKM | Fixed patch size |
| *Turbulent Radiative Layer 2D* | 3072 | **0.180** | 0.187 | 0.190 |
| | 768 | 0.210 | 0.210 | 0.210 |
| | 192 | 0.245 | 0.254 | 0.26 |
| *Active Matter* | 4096 | **0.0400** | **0.0405** | 0.0409 |
| | 1024 | 0.050 | 0.046 | **0.043** |
| | 256 | 0.066 | 0.070 | **0.050** |
| *Rayleigh-Bénard* | 4096 | **0.044** | 0.046 | 0.061 |
| | 1024 | **0.060** | **0.060** | 0.073 |
| | 256 | **0.074** | 0.080 | 0.083 |
| *Supernova Explosion* | 4096 | 0.270 | **0.258** | 0.261 |
| | 512 | 0.343 | **0.328** | 0.337 |
| | 64 | 0.370 | **0.364** | 0.367 |
| *Turbulence Gravity Cooling* | 4096 | 0.102 | **0.096** | 0.100 |
| | 512 | 0.152 | 0.138 | **0.133** |
| | 64 | 0.169 | 0.169 | **0.164** |

Table 12: Test VRMSE for different patch sizes (or, token counts) at inference time for three different 2D datasets using 7.6M parameter models. These values are for next-step prediction.

## I.2 ROBUSTNESS TO ADDING MORE PATCH/STRIDE OPTIONS

We performed additional experiments to explore the effect of adding more patch size options. Specifically, we ran our CKM + 50M axial ViT model with the following patch size sequences for the `turbulent_radiative_layer_2D` dataset (2, 3, 4 and 5 patch size choices). Note that we still choose patches as powers of 2, following standard practice in this field. We choose the following patch options:

$$[8, 16]; \quad [4, 8, 16]; \quad [4, 8, 16, 32]; \quad [4, 8, 16, 32, 64],$$

(also shown the fixed patch 16 as reference)

We report the next step VRMSE loss on the validation set for these four models when the inference is performed at a patch size of 16 for consistency across models. As detailed in the paper, note these are flexible CKM+axial ViT models which is trained only once, and then can be flexibly deployed at inference with multiple patch sizes.

| Patch sequence | VRMSE (Inference p.s.=16) |
|---|---|
| [16] – Fixed patch | 0.222 |
| [8, 16] | 0.208 |
| [4, 8, 16] | 0.204 |
| [4, 8, 16, 32] | 0.208 |
| [4, 8, 16, 32, 64] | 0.221 |

Based on the results above, the impact of adding additional patch sizes in this range is marginal, and for all of these models, the key conclusion of our paper holds: that we can achieve flexibility without a loss of accuracy over fixed patch baselines. We did not notice any training instabilities when increasing the number of patch sizes/strides shown during training. The loss curves remained smooth and convergent.

Additionally, as detailed in Appendix K, the PI-resize mechanism depends only on the base patch size and the randomly selected patch size, and the resulting pseudo-inverse matrix is a smooth function of these parameters. Therefore, introducing additional patch options does not introduce any inherent source of instability — an expectation that is fully supported by these empirical observations.

We additionally performed the same scaling experiment with the CSM+axial ViT model and obtained similar results. Below we show the next-step scaling VRMSE on the validation set. These new results show that the models are robust to adding more flexibility.

| Stride sequence | VRMSE (Inference stride=16) |
|---|---|
| [16] – Fixed patch | 0.222 |
| [8, 16] | 0.213 |
| [4, 8, 16] | 0.207 |
| [4, 8, 16, 32] | 0.209 |
| [4, 8, 16, 32, 64] | 0.216 |

Our method is robust to increased flexibility.

## I.3 UNDERLYING PATCH SIZE ABLATION IN CKM

We vary the underlying patch size of CKM (coupled with axial ViT). We use a sequence of base p.s. between 4, 8, 16 and 32. We tested on the *Turbulent Radiative Layer 2D* dataset.

As shown in Table 13, we find that all of the base patch sizes perform comparably to each other. Our conclusion on the adaptive compute, i.e., our claim that flexible models can provide compute adaptivity in PDE surrogates without compromising on the accuracy hold at all base patch size choices.

| Base patch size | Inference patch size | | |
| :---: | :---: | :---: | :---: |
| | 4 | 8 | 16 |
| 4 | 0.157 | 0.181 | 0.246 |
| 8 | 0.156 | 0.179 | 0.244 |
| 16 | 0.160 | 0.175 | 0.216 |
| 32 | 0.161 | 0.179 | 0.221 |

Table 13: Ablation study on the effect of the base patch size used in CKM, evaluated on the *Turbulent Radiative Layer 2D* dataset with an Axial ViT backbone. Values are for next-step prediction VRMSE.

### I.4 TIME CONTEXT LENGTH ABLATION

We kept the number of input time frames to be fixed at 6 because we wanted to isolate the effect of dynamic patching/striding in our model.

The choice of $n_{\text{time\_inputs}} = 6$ is consistent with other choices in the literature. We consider models that can produce stable long-horizon rollouts from a shorter input context stronger, since they learn to predict next steps with less ground-truth information. For reference, competitive PDE surrogate models such as CViT and SineNet use 10 input time steps, neural operator-based models like FNO, FFNO, and TFNO also commonly use 10, models such as MPP (NeurIPS 2024) and DISCO (ICML 2025) use 16. By using 6 input frames, our setup requires the model to predict longer rollouts from less context while still remaining broadly consistent with common practice in the literature.

We performed an ablation varying the number of input frames ($n_{\text{input\_steps}} = 3, 6, 12$) for our CKM + 50M AViT model. We kept all other settings and hyperparameters fixed. The results (next-step VRMSE on the validation set) are shown below:

| Input time steps | Next-step VRMSE |
| :---: | :---: |
| 3 | 0.202 |
| 6 (default) | 0.201 |
| 12 | 0.213 |

We find that varying the number of input steps in this range has only a marginal impact on accuracy. In practice, a smaller input context is also attractive for efficiency, since it reduces both training and inference cost while still providing stable long-horizon predictions.

### I.5 SENSITIVITY TO PATCH DIVERSITY DURING TRAINING

Next we performed an experiment to determine the effect of omitting patch sizes during training. To this end, we train on patch/stride sizes of 4 and 16 for our flexible models, and then during inference, we test on patch/stride of 4, 8 and 16. Clearly, as shown in Table 14, when we perform the evaluation on the omitted patch size of 8, the performance degrades for both CSM and CKM. This behavior is expected, and underlines the importance of training on a diverse patch/stride sizes to obtain the best generalization performance at inference.

| | Inference p.s.=4 | Inference p.s.=8 | Inference p.s.=16 |
| :--- | :---: | :---: | :---: |
| CKM + axial ViT | 0.158 | 0.453 | 0.230 |
| CSM + axial ViT | 0.160 | 0.377 | 0.222 |

Table 14: Performance drop when patch size 8 is omitted during training but used at inference. We trained models only on patch sizes 4 and 16. Results indicate the importance of patch diversity during training to ensure generalization at inference patch/stride choices.

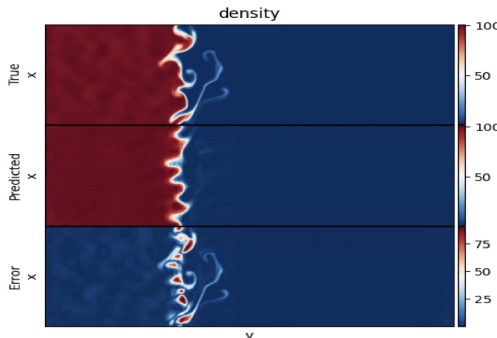

Figure 24: Flexified AFNO model with cyclic patches reduced artifacts, compared to Fig. 15

## J    NEURAL OPERATOR BASED NON-TRANSFORMER METHODS BENEFITING FROM SMALLER PATCH SIZES

In this work we focused on patch-based transformer architectures because they currently cover a broad range of competitive state-of-the-art PDE surrogate models (e.g., CViT (ICLR 2025), MPP (NeurIPS 2024), DISCO (ICML 2025), and POSEIDON (NeurIPS 2024)). However, non-transformer models such as AFNO (Guibas et al., 2021) share a similar motivation, and patch size turns out to be an important parameter for them as well.

To demonstrate this, we trained the AFNO baseline provided with the Well package on the *Turbulent Radiative Layer 2D* dataset with patch sizes of 16 and 4. Reducing AFNO's patch size from 16 to 4 improved the test VRMSE from 0.298 to 0.211, showing that patch size is a critical design choice. We also found patch size to be crucial in a recent DPOT (Hao et al., 2024) model, which performed large-scale pretraining on PDEs. These results highlight that patch size matters not only for transformer frameworks but also for other neural operator architectures.

To motivate this direction further, we ran the 20M-parameter fixed-patch-16 AFNO baseline (as provided in the Well package) on three 2D datasets and compared it with our flexible models and other baselines:

| Model / Dataset | Shear_flow | TRL_2D | Active_matter |
|---|---|---|---|
| AFNO (20M, p=16) | 0.689 | 0.486 | 0.990 |
| FFNO (7M) | 0.110 | 0.485 | 0.567 |
| SineNet (35M) | 0.170 | 0.650 | 0.760 |
| CSM + ViT (7M) | 0.0762 | 0.458 | 0.563 |
| CKM + ViT (7M) | 0.096 | 0.477 | 0.522 |

Table 15: Performance of AFNO and other baselines on three 2D datasets.

These benchmarks show that fixed-patch AFNO performs significantly worse than our flexified transformer-based models, especially on *Shear Flow* and *Active Matter*. We also tested a "flexified" AFNO by adding the CKM module to its patch embedding and de-embedding steps. On *Turbulent Radiative Layer 2D*, where AFNO was most competitive, the fixed-patch AFNO (p=16) reached a 10-step VRMSE of 0.486 with visible checkerboard artifacts. The CKM-flexified AFNO with alternating patch sizes reduced the VRMSE to 0.460 and mitigated the artifacts as shown below in Figure. 24, to be compared to artifact filled AFNO model in fig. 15.

Regarding DPOT, we note that CViT (which we have extensively studied in this paper) already provides detailed comparisons to DPOT in its original publication, and CViT outperformed DPOT across diverse benchmarks. We therefore chose CViT as the representative strong baseline for our experiments. These experiments confirm that researchers can incorporate flexible methods into architectures beyond transformers in a straightforward manner, enabling task-aware, inference-time control in both pretrained and task-specific surrogate models.

# K  DERIVATION OF THE PI-RESIZE MATRIX FOR CONVOLUTIONAL KERNEL MODULATORS

## K.1  MOTIVATION FOR KERNEL RESIZING

The CKM-based model dynamically adjusts convolutional kernel sizes at each forward pass while maintaining a fixed architecture. This allows variable patch sizes $(4, 8, 16)$, which is crucial for handling different spatial resolutions while ensuring compatibility with the ViT.

However, this flexibility introduces a problem: we train the CNN encoder, which extracts ViT tokens, with a *fixed base convolutional kernel*. To ensure that features extracted at different patch sizes remain aligned, we use a **resize transformation**, which projects the base kernel to a dynamically selected kernel size. Examples of resizing kernels through interpolation include (Xu et al., 2014) and (Beyer et al., 2023).

## K.2  MATHEMATICAL FORMULATION

Let:

- $W^{\text{base}} \in \mathbb{R}^{k^{\text{base}} \times k^{\text{base}} \times c_{\text{in}} \times c_{\text{out}}}$ be the learned CNN weights for the **fixed base kernel size** $k^{\text{base}}$.
- $W \in \mathbb{R}^{k \times k \times c_{\text{in}} \times c_{\text{out}}}$ be the resized kernel for a dynamically selected kernel size $k$.
- $B$ be an **interpolation matrix** (e.g., bilinear, bicubic) that resizes **input patches** from size $k$ to size $k^{\text{base}}$.
- $x \in \mathbb{R}^{B \times k \times k \times c_{\text{in}}}$ be an **input patch**, where:
  - $B$ is the batch size.
  - $k \times k$ is the **local receptive field** defined by the kernel.
  - $c_{\text{in}}$ is the number of input channels.

Intuitively, the goal is to find a new set of patch- embedding weights $W$ such that the tokens of the resized patch match the tokens of the original patch

$$W^{\text{base}} * x \approx W * (Bx), \tag{10}$$

where $*$ denotes convolution. The goal is to solve for $W$. Mathematically, this is an optimization problem.

## K.3  LEAST-SQUARES OPTIMIZATION FOR PI-RESIZE

### K.3.1  EXPANDING THE OBJECTIVE

We aim to expand the expectation:

$$\mathbb{E}_{x \sim \mathcal{X}} \left[ (x^T W^{\text{base}} - x^T B^T W)^2 \right]. \tag{11}$$

Since inner products can be rewritten as matrix-vector multiplications, we note that:

$$\langle x, W^{\text{base}} \rangle = x^T W^{\text{base}}.$$

**Step 1: Expand the Square**   Using the identity $(a - b)^2 = a^2 - 2ab + b^2$, we expand:

$$(x^T W^{\text{base}} - x^T B^T W)^2 = (x^T W^{\text{base}})^2 - 2x^T W^{\text{base}}(x^T B^T W) + (x^T B^T W)^2. \tag{12}$$

**Step 2: Take the Expectation**   Now, we take expectation over $x \sim \mathcal{X}$:

$$\mathbb{E}_{x \sim \mathcal{X}} \left[ (x^T W^{\text{base}})^2 - 2x^T W^{\text{base}}(x^T B^T W) + (x^T B^T W)^2 \right]. \tag{13}$$

Using the definition of the covariance matrix:

$$\Sigma = \mathbb{E}_{x \sim \mathcal{X}}[xx^T], \tag{14}$$

we apply the linearity of expectation to each term:

$$\mathbb{E}_x[(x^T W^{\text{base}})^2] = W^{\text{base},T} \Sigma W^{\text{base}}, \tag{15}$$

$$\mathbb{E}_x[(x^T B^T W)^2] = W^T B \Sigma B^T W, \tag{16}$$

$$\mathbb{E}_x[x^T W^{\text{base}}(x^T B^T W)] = W^{\text{base},T} \Sigma B^T W. \tag{17}$$

**Step 3: Write in Matrix Form**    Substituting these into the expectation:

$$\mathbb{E}_{x \sim \mathcal{X}}\left[(x^T W^{\text{base}} - x^T B^T W)^2\right] = W^{\text{base},T} \Sigma W^{\text{base}} - 2W^{\text{base},T} \Sigma B^T W + W^T B \Sigma B^T W. \tag{18}$$

The above expression can be re-written as:

$$\|W^{base} - B^T W\|_\Sigma^2 = (W^{base,T} - W^T B)\Sigma(W^{base} - B^T W). \tag{19}$$

**Conclusion**    This result expresses the squared error between the transformed weight matrix $W$ and the resized weight matrix $B^T W^{\text{base}}$, weighted by the covariance matrix $\Sigma$. The quadratic form measures the deviation in terms of feature space alignment, ensuring that the transformation remains consistent with the learned base kernel.

The optimal solution minimizing (19) is given by

$$W = \left(\sqrt{\Sigma} B^T\right)^\dagger \sqrt{\Sigma} W^{base} \tag{20}$$

where $\left(\sqrt{\Sigma} B^T\right)^\dagger \sqrt{\Sigma}$ is the Moore-Penrose pseudoinverse matrix.

### K.4    FINAL TRANSFORMATION AND IMPLEMENTATION

Thus, at each forward pass, we apply the following steps:

1. **Randomly select a kernel size** $k \in \{4, 8, 16\}$.
2. **Compute the interpolation matrix** $B$ that resizes the input patch from $k$ to $k^{\text{base}}$.
3. **Transform the kernel weights** using:

$$W = (B^T)^\dagger W^{\text{base}}. \tag{21}$$

4. **Apply the resized kernel** $W$ for convolution.

This ensures that dynamically changing kernel sizes maintain compatibility with the fixed architecture while enabling flexible patch sizes.

### K.5    CSM PSEUDOCODE

---

**Algorithm 1** Convolutional Stride Modulator (CSM)

---

**Input:** $x \in \mathbb{R}^{B \times H \times W \times T \times C}$
**Output:** $\hat{x} \in \mathbb{R}^{B \times H \times W \times 1 \times C}$
$k^{\mathtt{base}}$: fixed kernel size

**Step 1: Padding**
- Pad $x$ with learned tokens based on boundary conditions.

**Step 2: Stride selection**
- Sample stride(s) $s$ from $\{4, 8, 16\}$.
- If the encoder/decoder is multi-stage, split $s$ across stages.

**Step 3: Encoding**
1. For each encoder stage with assigned $s$:
   - $x \leftarrow \mathtt{Conv}(x, k^{\mathrm{base}}, \mathtt{stride} = s)$

**Step 4: Transformer processing**
- Pass tokens through transformer processor (architecture-agnostic)

**Step 5: Decoding**
1. For each decoder stage with assigned $s$:
   - $\hat{x} \leftarrow \mathtt{ConvTranspose}(x, k^{\mathrm{base}}, \mathtt{stride} = s)$

**return** $\hat{x}$

---

## K.6 CKM PSEUDOCODE

---

**Algorithm 2** Convolutional Kernel Modulator (CKM)

---

**Input:** $x \in \mathbb{R}^{B \times H \times W \times T \times C}$
**Output:** $\hat{x} \in \mathbb{R}^{B \times H \times W \times 1 \times C}$
`w`$^{\text{base}}$`:` base kernel weights, `B:` resizing matrix

**Step 1: Sample patch size**
- Sample patch size $k \in \{4, 8, 16\}$ once per forward pass.
- If the base encoder/decoder to be flexified is multi-staged, split $k$ across stages.

**Step 2: Encoding**
1. For each encoder stage with assigned $k$:
    - Resize kernel: $w \leftarrow B^{T\dagger} w^{\text{base}}$
    - $x \leftarrow \texttt{Conv}(x, w, \texttt{stride} = k)$

**Step 3: Transformer processing**
- Pass tokens through any transformer processor (architecture-agnostic)

**Step 4: Decoding**
1. For each decoder stage with assigned $k$:
    - Resize kernel: $w \leftarrow B^{T\dagger} w^{\text{base}}$
    - $\hat{x} \leftarrow \texttt{ConvTranspose}(x, w, \texttt{stride} = k)$

**return** $\hat{x}$

---

LLM USAGE

We used large language models (LLMs) during this project. We employed them as general-purpose assistive tools, comparable to an IDE or grammar-checking software. Specifically, we used them for (i) code assistance, including generating boilerplate, debugging, and suggesting implementation details, and (ii) language editing, including grammar, clarity, and stylistic improvements to drafts. All generated content was reviewed, corrected, and verified by the authors, who take full responsibility for the final text and code.

