# OpenReview forum: "Overtone: Cyclic Patch Modulation for Clean, Efficient, and Flexible Physics Emulators"
_ICLR.cc/2026/Conference — ICLR 2026 Poster_

### Official Review · Reviewer_2Q4X · 2025-10-28

**Soundness:** 3
**Presentation:** 2
**Contribution:** 3
**Rating:** 4
**Confidence:** 4

**Summary:**

This paper, "Overtone: Cyclic Patch Modulation for Cleaner, Faster Physics Emulators," introduces a novel framework named Overtone to address two critical challenges in Transformer-based partial differential equation (PDE) surrogate models during autoregressive rollouts: harmonic error accumulation caused by fixed patch sizes and the inflexibility of computational cost. The authors propose two architecture-agnostic modules—CSM (Convolutional Stride Modulation) and CKM (Convolutional Kernel Modulation)—that dynamically cycle through different patch sizes during inference. This approach effectively suppresses harmonic artifacts while enabling compute-adaptive deployment. Extensive experiments on various 2D and 3D PDE benchmarks demonstrate significant error reduction and flexible accuracy-speed trade-offs.

**Strengths:**

1. **Well-Defined and Significant Problem:** The paper accurately identifies a fundamental limitation in current Transformer-based PDE surrogates: systematic harmonic error accumulation due to fixed patch sizes. This issue is crucial for the long-term stability of rollouts.
2. **Novelty of Inference-Time Cyclic Modulation:** The proposed strategy of cyclically modulating patch sizes during inference is pioneering in this field. It offers a fresh perspective on mitigating harmonic artifacts while simultaneously achieving computational adaptability.
3. **Theoretical Underpinnings:** The mathematical analysis using a linearized error model provides theoretical support for the observed harmonic error accumulation, enhancing the rigor of the work.
4. **Comprehensive and Solid Experimental Design:** The experimental section is thorough, encompassing multiple PDE datasets and model architectures. Extensive ablation studies robustly validate the method's effectiveness and generalizability.

**Weaknesses:**

1. **Insufficient Justification for the Training-Inference Strategy Discrepancy:** A notable and somewhat counterintuitive finding is the requirement for a cyclic strategy during inference, despite using random patch size sampling during training. The observation that a random schedule significantly harms inference performance lacks deep theoretical explanation and experimental substantiation:
   - The current explanation, primarily based on a brief comparison in Appendix H.6, states that random scheduling hurts performance but does not adequately elucidate **why randomness is beneficial during training yet detrimental during inference**.
   - There is a lack of in-depth investigation into the **optimality of the cyclic strategy**. Why was the specific sequence `[4, 8, 16]` chosen? How would other deterministic sequences—such as non-monotonic patterns (e.g., `[4, 16, 8]`) or sequences with different periodicities (e.g., `[4, 8, 8, 16, 8, 4]`)—perform? A more systematic exploration of alternative cyclic schedules is needed.
2. **Investigate the Impact of Training Strategy on Artifacts and Method Efficacy**:
   - Clearly specify the training methodology used (e.g., teacher forcing vs. mixed teacher forcing/rollout training).
   - If only teacher forcing was used, discuss the potential impact of rollout training on the emergence of harmonic artifacts and verify whether the proposed cyclic modulation remains beneficial in that regime. This would strengthen the claims regarding the method's robustness and practical utility.

**Questions:**

See Weaknesses

---

### Official Review · Reviewer_u9p5 · 2025-10-28

**Soundness:** 2
**Presentation:** 3
**Contribution:** 2
**Rating:** 2
**Confidence:** 3

**Summary:**

This paper focuses on the systematic error accumulation during the autoregressive prediction of the fixed patch vision transformer. The authors observe that cyclically modulating patch sizes during autoregressive rollouts distributes errors across the frequency spectrum. Specifically, Overtone is presented, which contains dynamic stride modulation (CSM) and dynamic kernel resizing (CKM) to implement the cyclic patch modulation. As a result, Overtone achieves 40% error reduction in long rollouts.

**Strengths:**

-	The authors notice and theoretically analyze an interesting problem, that is, systematic harmonic artifacts in the rollout of fixed patch vision transformers.

-	The proposed method is verified to be effective in long-term rollout.

-	Sufficient implementation details are included.

**Weaknesses:**

Despite the above strengths, I think this paper has some implementation errors, which may cause the experimental results to be meaningless.

### (1) Potential implementation error.

In canonical vision transformer (ViT) and its follow-ups, such as Swin Transformer, patch embedding is implemented as a flattening and linear projection. In contrast, this paper adopts convolutional layers for patch embedding and decoding. I do not think this is a correct implementation. Thus, I think the proposed method may fail in handling canonical vision transformers.

### (2) A missing simple baseline.

In video prediction, a simple baseline to reduce such autoregressive error is rollout training. I think in the current experiments, all the models are trained in teacher forcing, which means during training, all the model input is ground truth. In such teacher-forced training, it is easy to observe so-called “systematic harmonic artifacts”. However, I doubt that such an accumulation error will be significantly resolved if the authors apply rollout training. Even with rollout training for about 10 steps, the model will be significantly improved.

Thus, I suggest that the authors rerun the experiments under a rollout training configuration. At least, provide some investigations into the effect of rollout training.

### (3) More Transformer baselines.

Additionally, I think some transformer-based models should also be compared, such as [1,2].

[1] Scalable Transformer for PDE Surrogate Modeling, NeurIPS 2023

[2] Unisolver: PDE-Conditional Transformers Are Universal PDE Solvers, ICML 2025

### (4) Missing comparison with the ground truth spectrum.

I appreciate that the authors provide the spectrum comparison in Figure 2 and the visualization comparison in Figures 4 and 5. However, I think Figure 2 should also involve the spectrum of ground truth, and Figures 4 and 5 are also expected to record the spectrum. I think this will make the comparison more direct and intuitive. Additionally, since the authors mentioned that “Overtone distributes errors across the frequency spectrum, preventing the systematic harmonic artifacts that plague fixed-patch models”, I am curious about the gap between the distributed spectrum and the ground truth.

### (5) Limitation in irregular geometries.

All the patch-based transformers are limited to regular geometries. I think the authors should discuss this in the limitations section.

**Questions:**

(1)	How to decide the hyperparameter in CSM and CKM, such as the adaptive patch size or stride size, as well as the cycle size.

(2)	It seems that Transolver performs quite badly. Do you follow their official configuration for the Navier-Stokes benchmark?

(3)	In Figure 4, although both CSM and CKM do not contain the harmonic artifacts, their predictions are still different from the ground truth. Do you have any interpretation about this?

(4)	How much extra computation overload do you need in training CSM and CKM?

---

### Official Review · Reviewer_wD4o · 2025-10-31

**Soundness:** 3
**Presentation:** 3
**Contribution:** 2
**Rating:** 4
**Confidence:** 4

**Summary:**

The paper identifies a key failure mode in Transformer-based PDE surrogates: the use of fixed patch sizes in autoregressive rollouts leads to the systematic accumulation of errors at specific harmonic frequencies, manifesting as grid-like visual artifacts. The diagnosis of temporally coherent harmonic error accumulation is a valuable insight. The proposed method, implemented via two architecture-agnostic modules (CSM and CKM), not only improves the stability and accuracy of long-horizon predictions but also offers a practical way to trade computational cost for precision at inference time without retraining.

Despite the good empirical results, the paper is weakened by a lack of deep theoretical grounding and leaves several critical questions about the method's underlying mechanics unanswered.

**Strengths:**

*   The paper's core strength lies in identifying and articulating the problem of harmonic error accumulation due to temporal coherence. This is a subtle but important issue that has been overlooked. The proposed cyclic modulation strategy is a simple and effective solution that directly addresses this root cause, and the empirical results demonstrate its success.

*   The authors conduct a thorough and rigorous experimental evaluation. The use of diverse and challenging 2D/3D datasets from The Well benchmark ensures the findings are relevant. The comparison against not only fixed-patch baselines but also a range of other non-patch-based models (FFNO, SineNet, etc.) provides a robust contextualization of the method's performance.

*   The Overtone framework is designed to be architecture-agnostic, a claim supported by its successful application to vanilla ViT, Axial ViT, and CViT.

**Weaknesses:**

While the method is empirically successful, there are several areas of concern that detract from the paper's overall quality.

*  The theoretical analysis in Section 2 and Appendix A provides some solid insights. But I don't quite understand how changing patch grid temporally thins and phase-misaligns injections which then shifts a portion of the error growth from quadratic to linear. (line 847~852)

*  A major concern is the physical intuition behind cyclically modulating the input representation over time. I agree on the author's claim, that when the patch grid is fixed, the same locations see repeated errors step after step, causing these errors to reinforce and appear as grid-aligned patterns or “checkerboards” over long horizons. I also believe, at each autoregressive step, changing the patch size effectively changes the model's "receptive field", which helps improve the performance. But what doesn't make sense to me is when it's applied temporally like in this paper. This raises a fundamental question: **why should the model's perception of the physical state oscillate in this deterministic, cyclic manner?** This seems physically implausible. Furthermore, it's unclear on an implementation detail: **how are token embeddings handled with the changing patch size?** In a standard ViT, the embedding dimension is fixed to 786 as it's $c \cdot p^2$. If the embedding dimension is kept fixed while the patch size changes, the model is cyclically fed tokens that represent vastly different amounts of spatial information. The rationale for why this is beneficial *temporally* is not well-explained.

*  Missing Critical Ablations and Comparisons:
    1.  The authors claim their method is architecture-agnostic and show that Swin Transformers suffer from similar artifacts (Appendix E). However, they do not present results for applying their method to a Swin Transformer. This is a logical omission that weakens the generality claim.
    2.  The benefit of *temporal cycling* is not de-confounded from the benefit of simply having a model that is purely *spatially cycling*. An alternative model design could process the entire input with three different patch sizes (using three separate but smaller models) in every time step. Without this comparison, it's unclear if the cyclic schedule is the key ingredient, or if the training on diverse patch sizes is doing most of the work.


*   The evaluation relies solely on a single error metric (VRMSE). For physics simulation tasks, it is crucial to analyze whether the model preserves important derived physical quantities. For example, in the fluid dynamics datasets, an analysis of the predicted vorticity field (derived from the velocity) would provide much deeper insight into whether the model is learning the correct physical structures or merely achieving low pixel-wise error.

*  There is no shape information in figure 3. You can just use same shape in green.

**Questions:**

See questions from weaknesses above.

---

### Official Review · Reviewer_M73p · 2025-11-01

**Soundness:** 2
**Presentation:** 2
**Contribution:** 2
**Rating:** 2
**Confidence:** 4

**Summary:**

This paper introduces Overtone, a framework for transformer-based PDE surrogate models. The model is trained once using randomly sampled patch sizes or strides. During inference, this single model can be deployed in two modes: (1) at a fixed patch size, which allows users to dynamically trade computational cost for accuracy, or (2) using a cyclic modulation of patch sizes during autoregressive rollouts. The authors evaluate this framework on 2D and 3D PDE benchmarks from *The Well* collection, primarily comparing against multiple, individually trained, fixed-patch-size baselines, as well as other SOTA surrogates such as FFNO and SineNet. The results indicate that the single Overtone model matches or exceeds the performance of other models, and that the cyclic rollout strategy mitigates harmonic error accumulation, leading to improved accuracy in long-horizon predictions.

**Strengths:**

- The authors analyzed the phenomenon in Transformer-based PDE prediction where fixed patch grids lead to error accumulation at specific harmonic frequencies, which manifests as spatial artifacts. To address this, they adopt strategies common in computer vision, incorporating multi-size kernels and multi-strides within a single model, thereby mitigating this problem in ViTs and achieving modest performance improvements.

- The authors conduct evaluations on multiple PDE datasets from *The Well* benchmark and provide comparisons against several non-Transformer-based Neural Operators.

**Weaknesses:**

- Insufficient Baselines: The paper claims flexibility but lacks critical comparisons against existing ViT variants that also process multi-scale information (e.g., U-ViT or Swin-Unet), making Overtone's relative advantages unclear. The authors are expected to add direct performance comparisons against these relevant baselines in their revision.

- Limited Novelty: The core modules (CKM and CSM) are not novel. CKM is heavily based on prior work such as FlexiViT, and CSM (stride modulation) is a standard technique, limiting its contribution to an application of known methods to autoregressive PDE rollouts.

- Presentation: There are some problems with the layout of the charts in the text, and the authors are expected to refine them during revisions.

**Questions:**

See weaknesses

---

### Author Response · Authors · 2025-12-03
**Combined rebuttal, part 1**

## To the Area Chair and Reviewers

We thank all reviewers for their time and detailed comments. Several reviews highlight the importance of understanding harmonic artifacts in patch-based PDE surrogates and find our cyclic patch modulation both simple and practically useful. In this response we clarify two aspects that are central to the paper but not fully emphasised in all reviews: (i) the novelty of our inference-time cyclic patch/stride modulation for PDE surrogates, and (ii) the strength and breadth of the empirical evaluation on The Well benchmarks.

## Reviewer M73p

**Novelty**

We disagree that our work has "limited novelty." Our central contribution is *not* just re-using multi-scale kernels. It is rather a new inference-time rollout strategy for Transformer PDE surrogates: deterministic cyclic modulation of patch sizes/strides across autoregressive steps, motivated by a linearized error analysis that explains how fixed patch grids create temporally coherent harmonic artifacts. Modulating patch/stride *over time* in this way has never been used in either PDE surrogate or vision literature. Note that the effect of this cycling has a theoretically-motivated and empirically observed effect on error.

CKM/CSM are deliberately simple so that (i) the novelty is clearly in the rollout scheme and error analysis, and (ii) the mechanism can wrap standard ViT-style backbones. This is reflected in other reviews, which explicitly call our inference-time cyclic modulation "pioneering" and emphasize the value of the theoretical error analysis.

**Baselines and architectural comparisons**

Our aim is to isolate the *effect of patch/stride modulation and compute-adaptive inference*, not to exhaustively benchmark all Transformer variants. We therefore:

-   Compare against strong, widely used PDE surrogates (TFNO, FFNO, Transolver, SineNet) and show our flexible models are competitive or substantially better on challenging Well benchmarks.

-   Compare a single CKM/CSM model against several fixed-patch ViTs trained at patch sizes 4, 8, 16 under the *same total training compute*.

Architectures such as U-ViT and Swin-UNet modify the backbone; they are orthogonal to our contribution, which is a wrapper that could in principle be applied on top of such models in future work. Their absence does not change the main empirical finding: a *single* compute-elastic model can match or beat multiple static models while also enabling a principled cyclic rollout strategy.

**Magnitude of improvements**

The gains are not "modest." Across multiple 2D and 3D PDE benchmarks from The Well, cyclic rollouts reduce long-horizon error by up to \~40% relative to fixed-patch baselines and visibly suppress grid-locked artifacts. For long rollouts on complex turbulent systems, this is a substantial effect.

**Presentation**

The review mentions "problems with layout of charts" but does not specify issues. Plots such as Figure 3 already include the key information needed to assess the compute-accuracy trade-off, with three fixed-patch baselines and our flexible models across token counts on multiple datasets. Any minor cosmetic issues (e.g., adding explicit shape annotations) are straightforward clarifications and do not impact the technical content or conclusions.

---

### Author Response · Authors · 2025-12-03
**Combined rebuttal, part 2**

## Reviewer wD4o

We appreciate the recognition of the importance of harmonic error accumulation and the thoroughness of the experiments.

**(1) Theory: quadratic vs. linear error growth**

In our linearized analysis, the expected error energy at a frequency omega after n steps decomposes into: (i) a term that is always linear in n, and (ii) a correlation term that depends on how phase-aligned error injections are across timesteps. For a fixed patch grid, on problematic harmonics the correlation term behaves like a sum 1+2+...+n, i.e. O(n^2), and dominates. With temporal cycling, the patch grid changes so that injections are typically *not* phase-aligned, driving the correlation term toward zero and leaving the linear term dominant. This is the sense in which cycling shifts part of the growth from quadratic to linear, consistent with the observed 40% rollout error reduction.

We agree this analysis is approximate: the contribution is primarily empirical, and the theory is meant as a physically motivated explanation rather than a full proof.

**(2) Physical intuition and token embeddings**

Implementation-wise, the embedding dimension is fixed as in a standard ViT. Changing patch size only alters (1) the number of tokens and (2) the spatial footprint per token. During training we randomize over patch/stride sizes {4, 8, 16}, so the network explicitly learns to interpret tokens from all these grids within a single shared embedding space. Small patches capture fine-scale structure; larger patches emphasize coarse/global structure. Alternating them over time lets the model repeatedly "refresh" fine and coarse scales, which empirically reduces checkerboard artifacts and improves long-horizon accuracy.

So while a deterministic schedule may look "physically odd" at first glance, it is simply a structured way of varying the receptive field of an already multi-patch-trained model, not arbitrarily oscillating the physical representation.

**(3) Swin, "purely spatial cycling," and metrics**

-   **Swin + Overtone**: Our "architecture-agnostic" claim means CKM/CSM assume only a patch/grid-based encoder-decoder, not a specific ViT. Appendix E includes Swin to show that *even Swin* exhibits fixed-grid artifacts once wrapped in a standard encoder/decoder. We did not add Swin+CKM/CSM purely due to compute cost of repeating the full experimental suite on all datasets and architectures, but we expect our mechanism to transfer.

-   **"Purely spatially cycling" baseline**: Cycling between three separately trained fixed-patch models at each step is a *different design goal*: it requires training and maintaining multiple networks and does not provide a single compute-elastic surrogate. Our goal is to answer a practical question about whether one flexible model can replace several static ones under the same training budget, so we explicitly match total training compute between (i) three fixed-patch models and (ii) one CKM/CSM model. Under this constraint, our flexible model matches or outperforms the static baselines, which directly supports the paper's main claim.

-   **VRMSE vs. physical diagnostics**: We use VRMSE because it is a de-facto standard pointwise metric in PDE surrogate work and provides a clear compute-accuracy trade-off. However, we *do* also probe physical consistency: Appendix G evaluates mass and momentum conservation on the shear-flow dataset over 50 rollout steps and shows normalized deviations <=1% for momentum and \<10% for mass, remaining bounded over long horizons. This indicates that Overtone's improvements do not come at the expense of basic physical structure.

---

### Author Response · Authors · 2025-12-03
**Combined rebuttal, part 3**

## Reviewer u9p5

We appreciate the recognition of the identified harmonic artifact problem and the strong long-rollout performance.

**(1) Alleged implementation error (convolutional patch embedding)**

We strongly disagree that our use of convolutional layers for patch embedding/decoding is an "implementation error." A convolution with kernel size and stride equal to the patch size is mathematically equivalent to "flatten + linear projection" and is in fact the *standard* implementation in many ViT variants for efficiency. This design is used both in PDE models (e.g., MPP, Poseidon, CViT, AFNO, DPOT) and in vision ViTs like FlexiViT and CvT. Our method therefore follows common practice and remains compatible with canonical ViT architectures.

**(2) Rollout training vs. teacher forcing**

All models, including baselines and Overtone, are trained under teacher forcing, which is the default in The Well and in many PDE surrogates. This keeps the comparison controlled: any differences can be attributed to the patch/stride strategy rather than to differing training setups.

Rollout training is complementary to our contribution. It could in principle lower absolute error for *all* models, including ours. But the failure mode we target, grid-locked harmonic accumulation from a fixed patch grid, is a property of the rollout geometry, not of teacher forcing per se, so we expect cyclic modulation to remain beneficial even under rollout training. A full exploration of rollout training on all high-resolution Well datasets (>=128^2, 64^3) would be substantially more expensive and is outside our compute budget for this submission.

**(3) Additional Transformer baselines and spectra**

We already compare against strong non-patch PDE surrogates (TFNO, FFNO, Transolver, SineNet) and comprehensive fixed-patch ViT baselines; these are the most relevant competitors for our compute-adaptive tokenization and cyclic rollout mechanism. Models like Scalable Transformer and Unisolver are interesting but change the backbone rather than the rollout strategy; they are orthogonal to our contribution and would not alter the core message that cyclic modulation in a single flexible model improves long-horizon performance and compute flexibility.

Regarding spectra: Figure 2 plots the *residual* spectrum (difference between prediction and ground truth). For the ground truth itself this residual is identically zero, so a separate "ground-truth spectrum" curve would be trivial and is omitted.

We agree that patch-based models are limited to regular grids and already note this limitation; this is a general constraint of grid-tokenized Transformers rather than a specific flaw of Overtone.

**(4) Hyperparameters, Transolver configuration, and compute**

-   **Hyperparameters / cycle size**: Training-time patch sets are chosen from standard values used in PDE ViT work ({4, 8, 16}), and Appendix H shows that adding larger patches (32, 64) keeps the method robust while increasing patch diversity.

-   **Transolver performance**: We follow the official Transolver configuration (8 layers, 8 heads, 256 channels, etc.) at higher Well resolutions (>=128^2) than the original 64x64 Navier-Stokes and 85x85 Darcy setups, which makes the task harder; despite tuning learning rates we find Transolver struggles on these regimes.

-   **Training compute**: As noted above, we explicitly match the total training budget between three fixed-patch models and a single CKM/CSM model; the flexible model therefore does *not* require extra training compute beyond what is already spent on the static baselines.

---

### Author Response · Authors · 2025-12-03
**Combined rebuttal, part 4**

## Reviewer 2Q4X

We are grateful for this reviewer's positive assessment of the problem, novelty, theory, and experimental design.

**(1) Training-inference discrepancy and randomness vs. cyclic schedule**

**Why random patch sampling during training?** During training we treat patch/stride size as data augmentation and capacity sharing: sampling p in {4, 8, 16} exposes a single model to all resolutions it will see at test time and teaches it to process all patch configurations consistently, analogous to noise injection in standard deep learning.

**Why a deterministic cycle at inference?** At inference we want (i) a simple, deterministic policy for users and (ii) a schedule aligned with the harmonic-error intuition. A random rollout keeps re-sampling patch sizes and can occasionally re-align grid boundaries, re-introducing localized build-up of error and adding variance across runs. Empirically, Appendix H.6 shows that the alternating/cyclic schedule consistently performs as well or better than randomized schedules across several datasets. Randomness is thus helpful when *learning* a multi-patch representation, but a structured schedule is preferable when *deploying* it.

**Optimality of \[4, 8, 16\]**

Our goal in this first work is not to claim \[4, 8, 16\] is optimal, but to demonstrate that even a simple deterministic schedule yields substantial gains. The choice is motivated by common patch sizes in PDE ViTs and by the requirement that they evenly divide the Well resolutions. Our analysis suggests two essential ingredients: (i) multiple patch sizes per period and (ii) avoiding long runs of a single size so the grid changes regularly. Exploring more exotic sequences is a natural direction for follow-up work rather than a prerequisite for the current claims.

**(2) Training strategy (teacher forcing vs. rollout training)**

We use teacher forcing throughout, which matches The Well setup and the majority of PDE surrogates we compare against. This ensures a fair comparison across all baselines.

The mechanism we target, the accumulation of grid-locked harmonic artifacts on a fixed patch grid, does not depend on teacher forcing specifically, so we expect cyclic modulation to remain beneficial under rollout training as well. A full study of rollout training on all high-resolution datasets would require significantly more compute and was therefore left as future work.

---

### Author Response · Authors · 2025-12-03
**Combined rebuttal, part 5/5**

## Summary for the AC

Across reviews, there is broad agreement that:

-   The paper identifies an important, previously under-analyzed failure mode of fixed-patch Transformer PDE surrogates.

-   The cyclic rollout mechanism is simple, effective, and practically useful, and the experiments on diverse Well benchmarks are thorough.

The main criticisms concern additional baselines, more exhaustive schedule explorations, and deeper theory, all valuable, but in our view beyond the scope of a single empirical paper that already introduces a new rollout strategy, a compute-elastic surrogate, and extensive ablations under a fixed compute budget.

Finally, we would like to highlight that several natural extensions of the current experiments (more backbones, schedule variants, rollout training, multiple seeds) are constrained by compute rather than by lack of interest. A single configuration in Table 2 already requires roughly 24 hours on 8xA100-80GB, i.e. about USD 350 at typical cloud pricing. The full table involves 72 such runs (note that each CKM/CSM model is trained once then evaluated at multiple token counts), which corresponds to around USD 25,000 of training compute alone, before adding a further ~20% for the long-horizon evaluation. Adding just two extra seeds per configuration would roughly triple this cost to ~USD 75,000.

Taken together with the factual misunderstandings we have highlighted above (for example, calling a standard ViT-style convolutional patch embedding an "implementation error," or describing up to \~40% long-horizon improvements as "modest") and the structural compute constraints outlined here, we believe that the two lowest reviews may not be fully reliable indicators of the quality or relevance of the work. We would encourage the AC to weigh them accordingly, in light of the more positive and technically engaged reviews that emphasize the novelty and practical impact of Overtone.

At this scale, it is standard in large PDE surrogate work not to provide full confidence intervals or exhaustive architectural sweeps; recent models such as Poseidon, DISCO and MPP make the same trade-off. In our view, several critical comments about "missing" baselines therefore reflect a mismatch between expectations and the practical compute budget for 100M-parameter physics emulators, rather than substantive weaknesses in the method itself. We hope this clarification helps the AC weigh the reviews and see the work as a solid and genuinely novel step for Transformer-based physics emulators.

---

### Meta-Review · Area_Chair_uLk6 · 2025-12-30

**Summary:**

I find this paper makes a good empirical contribution to ViT-based PDE surrogate modeling. The core insight—that fixed-patch tokenization causes systematic error accumulation at harmonic frequencies during autoregressive rollouts—is novel and well-demonstrated through spectral analysis. The proposed cyclic patch modulation strategy is simple yet effective, yielding up to 40% error reduction in long-horizon predictions across diverse 2D and 3D PDE benchmarks from The Well. I am persuaded that the combination of harmonic mitigation and compute-adaptive deployment (trading accuracy for speed at inference without retraining) provides genuine practical value.

The reviews were split, with two recommending rejection and two marginally below threshold. After careful examination, I find one rejection (Reviewer u9p5) contains a factual error—the claim that convolutional patch embedding is an "implementation error" is incorrect, as this approach is standard in many ViT variants including MPP, Poseidon, CViT, and FlexiViT. The other rejection (Reviewer M73p) raises valid concerns about component novelty but does not adequately engage with the paper's core contribution, which lies in the cyclic rollout strategy and harmonic error analysis rather than the individual modules. The two higher-scoring reviews correctly identify both the strengths (problem diagnosis, practical utility) and legitimate concerns (theoretical depth, missing ablations), and I weight these more heavily in my assessment. Overall, I believe that the method has merit, the empirical results are strong, but the paper would be substantially stronger if some theoretical depth was also provided.

**Reviewer Concerns:**

Regarding novelty, Reviewer M73p argues that CKM and CSM are based on existing techniques. This is partially correct—kernel interpolation and stride modulation are not new. However, the novelty lies in applying cyclic modulation during autoregressive rollouts to mitigate harmonic error accumulation, which has not been explored in PDE surrogates or video modeling. The rebuttal appropriately distinguishes the contribution from the implementation details.

Reviewer u9p5's claim that convolutional patch embedding is an implementation error is factually incorrect. Convolutions with kernel size equal to stride equal to patch size are mathematically equivalent to flatten-and-project and are used in canonical architectures (ViT, FlexiViT, CViT, MPP, DPOT). I discount this concern entirely.

The concern about rollout training (raised by u9p5 and 2Q4X) is valid—the paper does not empirically verify that harmonic artifacts persist under rollout training. The authors reasonably argue that teacher forcing is standard in The Well benchmarks and rollout training is complementary rather than contradictory to their contribution. This remains a direction for future work but does not invalidate the current results.

Reviewer wD4o's concern about physical intuition for cyclic modulation is partially addressed. The explanation that alternating patch sizes redistributes where boundary errors are introduced, preventing coherent spatial accumulation, is physically reasonable if not rigorously proven. The theoretical analysis is explicitly presented as heuristic motivation, which I find acceptable for a purely empirical paper.

The missing Swin+Overtone experiments and alternative baseline comparisons (U-ViT, Unisolver) are acknowledged as compute-limited. Given the reported experimental costs (approximately USD 25,000 for the current experiments), this is a reasonable constraint. The requested architectures would change the backbone rather than test the rollout strategy, making them complementary to the core contribution.

**Reviewer Scores:**

Reviewer M73p: 2 → 3-4. The rebuttal clarifies that novelty lies in the cyclic rollout strategy, not the modules themselves. The "modest improvements" characterization contradicts the 40% error reduction demonstrated. With discussion, this reviewer would likely acknowledge some merit in the inference-time contribution?

Reviewer wD4o: 4 → 4-5. Concerns about physical intuition and theoretical depth are partially addressed. The conservation law analysis in Appendix G addresses the VRMSE-only concern. This is the most balanced review and would likely remain at 4 or move to 5.

Reviewer u9p5: 2 → 4. The "implementation error" claim is factually incorrect and should be retracted. With this corrected, remaining concerns about rollout training and additional baselines are reasonable but not disqualifying. A substantial score increase would be appropriate.

Reviewer 2Q4X: 4 → 5. Concerns about training-inference discrepancy are well-addressed by the rebuttal (random for learning multi-patch representations, cyclic for deployment). Appendix H.6 empirically supports cyclic over random schedules.

---

### Decision · Program_Chairs · 2026-01-26

Accept (Poster)